# Key evolutionary events in the emergence of a globally disseminated, carbapenem resistant clone in the *Escherichia coli* ST410 lineage

Yu Feng [1,2,3,7], Lu Liu [1,2,3,7], Ji Lin [4], Ke Ma [1,2,3], Haiyan Long [1,2,3], Li Wei [4], Yi Xie[5], Alan McNally [6,7] & Zhiyong Zong [1,2,3,4]

There is an urgent need to understand the global epidemiological landscape of carbapenem-resistant *Escherichia coli* (CREC). Here we provide combined genomic and phenotypic characterization of the emergence of a CREC clone from the ST410 lineage. We show that the clone expands with a single plasmid, within which there is frequent switching of the carbapenemase gene type between $bla_{NDM}$ and $bla_{OXA-181}$ with no impact on plasmid stability or fitness. A search for clone-specific traits identified unique alleles of genes involved in adhesion and iron acquisition, which have been imported via recombination. These recombination-derived allelic switches had no impact on virulence in a simple infection model, but decreased efficiency in binding to abiotic surfaces and greatly enhanced fitness in iron limited conditions. Together our data show a footprint for evolution of a CREC clone, whereby recombination drives new alleles into the clone which provide a competitive advantage in colonizing mammalian hosts.

[1] Center of Infectious Diseases, West China Hospital, Sichuan University, Chengdu, China. [2] Division of Infectious Diseases, State Key Laboratory of Biotherapy, Chengdu, China. [3] Center for Pathogen Research, West China Hospital, Sichuan University, Chengdu, China. [4] Department of Infection Control, West China Hospital, Sichuan University, Chengdu, China. [5] Laboratory of Clinical Microbiology, Department of Laboratory Medicine, West China Hospital, Sichuan University, Chengdu, China. [6] Institute of Microbiology and Infection, College of Medical and Dental Sciences, University of Birmingham, Birmingham, UK. [7] These authors contributed equally: Yu Feng, Lu Liu, Alan McNally. Correspondence and requests for materials should be addressed to Z.Z. (email: zongzhiy@scu.edu.cn1)

*E*scherichia *coli*, a member of the *Enterobacteriaceae*, is a major human pathogen causing various infections ranging from intestinal disease and urinary tract infections to invasive bloodstream infections. Carbapenems such as ertapenem, imipenem, and meropenem are potent antimicrobial agents against the *Enterobacteriaceae* and have become the mainstream agents of choice to treat severe infections caused by *E. coli*. This is due to the near-ubiquitous carriage of extended spectrum β-lactamases (ESBL) in *E. coli* causing urine and bloodstream infections[1]. However, carbapenem-resistant *E. coli* (CREC) has emerged worldwide, representing a serious challenge for clinical management and public health[2]. Carbapenem resistance in *E. coli* is largely due to the production of carbapenem-hydrolyzing enzymes (carbapenemases)[3]. There are a variety of carbapenemases, and the most common ones observed in clinical bacterial strains include KPC (*K*lebsiella *p*neumoniae *c*arbapenemase), NDM (*N*ew *D*elhi *m*etallo-β-lactamase), OXA-48 (*oxa*cillinase-48), IMP (*im*ipenemase), and VIM (*V*erona *i*ntegron-encoded *m*etallo-β-lactamase)[3,4]. NDM appears to be particularly common in CREC[5].

In contrast to the well-studied carbapenem-resistant *Klebsiella pneumoniae* (CRKP), the clonal background of CREC is less well characterized including the transmission of CREC within and between hospitals. Studies of global *E. coli* isolate collections have shown that carbapenemase gene carriage is focused in strains belonging to lineages within the phylogroups A and B1 *E. coli*[6], classically considered to be non-pathogenic commensals[7]. Local genomic epidemiological studies, such as of CREC in China, have also led to the discovery of globally disseminated clones ST167 and ST617, both of which belong to phylogenetic group A[8]. A similar study in Scandinavia also resulted in the discovery of a globally disseminated CREC lineage, ST410 (ref. [9]). This lineage was similar to the global pandemic ESBL *E. coli* lineage ST131, in that a specific clone (B4/H24RxC) had arisen from the background population via acquisition of a resistance plasmid, in this instance the ST410 lineage containing an IncX3 plasmid carrying the $bla_{OXA-181}$ carbapenemase gene[9].

The evolutionary steps leading to the emergence of the *E. coli* ST131 lineage have been extensively reported[10–13]. However, the lack of global concerted genomic analyses of CREC means that our understanding of how potentially dominant CREC clones are evolving and emerging is lacking. Analysis of the ST167 and ST617 lineages showed some clear overlaps in evolutionary trajectory between these CREC clones and ST131 including mutations involved in host colonization and in intergenic regions associated with emergence of multi-drug resistant (MDR) plasmid-bearing clones[8], but there remains a need to determine if this pattern is common across emerging CREC clones. Here we utilize a province wide analysis of clinical CREC strains performed at West China Hospital to address this question. Between June 2016 and February 2017 all CREC collected from eight Sichuan hospitals were genome sequenced. The majority (60%) of strains belonged to ST410, ST167, and ST617. Analysis of the ST410 genomes and comparison against all publicly available ST410 genome sequences confirmed the presence of an MDR B4/H24RxC clone within ST410 globally disseminating either $bla_{NDM-5}$ or $bla_{OXA-181}$. Long-read sequencing revealed these carbapenemases are freely interchanging on an identical IncX3 plasmid. Genetic loci which discriminate the MDR clone from the rest of the ST410 lineage included anaerobic metabolism loci and intergenic regions, as shown for other MDR clones of *E. coli*, and unique sequence variants of the *fhu* iron acquisition operon, which confer an increased ability to scavenge iron. Together our data show a footprint for evolution of a CREC clone, whereby recombination drives new alleles into the clone which provide a competitive advantage in colonizing mammalian hosts. The importance of enhanced colonization capabilities in the evolution of MDR clones must be fully characterized and presents a possible new avenue for combatting CREC.

## Results

### ST410 is the most common circulating CREC lineage in Sichuan strains.
A total of 25 CREC strains were collected from eight hospitals (Supplementary Table 1) in Sichuan province, China, between June 2016 and February 2017. The strains were recovered from blood, sputum, urine, wound secretion, bile, pleural fluid, and ascites, suggesting that CREC is associated with various types of infections such as bloodstream infection, pneumonia, and urinary tract infection (Table 1). All CREC strains were resistant to imipenem (minimum inhibitory concentrations [MIC], 8 to >256 mg/l), meropenem (MIC, 32 to >256 mg/l), piperacillin/tazobactam, ceftazidime, and ceftazidime/avibactam but were susceptible to tigecycline (Table 2 and Supplementary Dataset 1). Most strains were resistant to ciprofloxacin (resistance rate, 96%), trimethoprim/sulfamethoxazole (88%), aztreonam (76%), and gentamicin (68%), while most were susceptible to aztreonam/avibactam (susceptible rate, 92%), colistin (88%), and amikacin (72%) (Table 2). All of the 25 CREC were subjected to short read whole-genome sequencing and antimicrobial resistance genes were identified based on their draft genome sequences. $bla_{NDM}$ was the only carbapenemase-encoding gene identified and was found in all 25 CREC strains (Supplementary Dataset 1). Four $bla_{NDM}$ variants were identified including $bla_{NDM-5}$ (the most common type, present in 21 strains), $bla_{NDM-1}$ (in two strains), $bla_{NDM-7}$, and $bla_{NDM-21}$ (each in one strain) (Table 1). Three colistin-resistant CREC carried the plasmid-borne colistin-resistance gene *mcr-1*, one of which also had another colistin-resistance gene *mcr-3* (Table 1). Seven amikacin-resistant strains, all of which exhibited high-level resistance to amikacin (MIC, >256 mg/l), had the 16S rRNA methylase gene *rmtB*, and one strain had another 16S rRNA methylase gene *armA* in addition to *rmtB*. The CREC strains belonged to 13 sequence types (STs), highlighting a heterogeneous clonal background. Three STs, ST167 ($n = 4$), ST617 ($n = 5$), and ST410 ($n = 6$), accounted for the majority (60%) of CREC, while there was only a single strain for the remaining 10 STs. The number of SNPs among ST167 and ST617 strains are shown in Supplementary Tables 2 and 3. As we have previously characterized ST167 and ST617 CREC[8]. we therefore focused on ST410, the common type, in this study.

### Most Sichuan ST410 CREC belong to the globally spread B4/H24RxC clone.
Strain 020001, the first ST410 strain isolated in our study, was also subjected to long-read whole-genome sequencing using MinION (the sequencing yield is listed in Supplementary Table 4) to obtain its complete genome sequence. A hybrid assembly of the genome sequence of 020001 revealed that the strain had a 4.9-Mb chromosome and six plasmids (Supplementary Table 5). The chromosome sequence of strain 020001 was then used as a reference for mapping. Two (strains 020026 and 020031) of the six strains were separated from each other by 17 core single-nucleotide polymorphisms (SNPs; Table 3), indicating a potential clonal spread. Given that these two strains were recovered from different hospitals, such an observation suggests recent inter-hospital movement of a common strain. Another two strains (strains 020129 and 020147) were 51 to 90 SNPs distant from the above two strains, suggesting relatively recent shared ancestry for the four strains (Table 3). The remaining two strains had >2500 SNPs between each other and any of the aforementioned four strains (Table 3). This suggests that the two remaining strains had no recent linkage with

**Table 1 CREC strains in this study**

| Strain[a] | Sample | Hospital[b] | ST | ST complex | NDM | *mcr* | SRR accession no· |
|-----------|--------|-------------|-----|------------|-----|-------|-------------------|
| 020068 | Sputum | MS | 101 | 101 | NDM-5 | *mcr-1* | SRR6474931 |
| 020022 | Urine | YB | 156 | 156 | NDM-5 | *mcr-1* | SRR6474926 |
| 020007 | Urine | ZG | 167 | 10 | NDM-5 | | SRR6474927 |
| 020016 | Sputum | MS | 167 | 10 | NDM-5 | | SRR6942786 |
| 020033 | Blood | WCH | 167 | 10 | NDM-5 | | SRR6942788 |
| 020076 | Wound | MY | 167 | 10 | NDM-7 | | SRR6942790 |
| 020123 | Wound | WCH | 206 | 206 | NDM-5 | *mcr-1, mcr-3* | SRR7026301 |
| 020005 | Bile | ZG | 359 | 101 | NDM-5 | | SRR6942791 |
| 020119 | Urine | WCH | 361 | 361 | NDM-1 | | SRR7026295 |
| 020001 | Blood | ZG | 410 | 23 | NDM-5 | | SRR6942789 |
| **020026** | **Sputum** | **LS** | **410** | **23** | **NDM-5** | | **SRR6942787** |
| 020031 | Blood | WCH | 410 | 23 | NDM-5 | | SRR7026311 |
| **020032** | **Blood** | **WCH** | **410** | **23** | **NDM-5** | | **SRR6942781** |
| **020129** | **Sputum** | **WCH** | **410** | **23** | **NDM-1** | | **SRR7026307** |
| **020147** | **Blood** | **LS** | **410** | **23** | **NDM-5** | | **SRR7026287** |
| 020004 | Sputum | ZG | 448 | 448 | NDM-5 | | SRR6942792 |
| 020023 | Urine | YB | 617 | 10 | NDM-21 | | SRR6442663 |
| 020044 | Pus | YB | 617 | 10 | NDM-5 | | SRR7026293 |
| 020085 | Blood | YB | 617 | 10 | NDM-5 | | SRR7026292 |
| 020141 | Blood | YB | 617 | 10 | NDM-5 | | SRR7026290 |
| 020149 | Pleural fluid | LS | 617 | 10 | NDM-5 | | SRR7026286 |
| 020028 | Blood | WCH | 3052 | 38 | NDM-5 | | SRR6942782 |
| 020088 | Sputum | CD6 | 6388 | 101 | NDM-5 | | SRR6942784 |
| 020066 | Urine | MS | 6823 | 196 | NDM-5 | | SRR6942785 |
| 020048 | Ascites | LE | 7019 | 11 | NDM-5 | | SRR7026291 |

[a]The strains are added WCHEC (if from West China Hospital) or SCEC (if from other hospitals) in the name in SRR database. ST410 strains of the B4/H24RxC clone are highlighted in bold. The number of SNPs among ST167 and ST617 strains are shown in Supplementary Tables 2 and 3. [b]Hospitals: CD6, The Sixth People's Hospital of Chengdu City; LE, The People's Hospital of Leshan City; LS, The First People's Hospital of Liangshan Yi Autonomous Prefecture; MS, Meishan Hospital of Traditional Chinese Medicine; MY, Mianyang Central Hospital; WCH, West China Hospital of Sichuan University; YB, The Second People's Hospital of Yibin City; ZG, The First People's Hospital of Zigong City

**Table 2 In vitro susceptibility of the 25 CREC isolates**

| Antimicrobial agents | MIC range | S (%) | I (%) | R (%) |
|---------------------|-----------|-------|-------|-------|
| Amikacin | 2->256 | 18 (72) | 0 | 7 (28) |
| Aztreonam | ≤0.5->256 | 2 (8) | 4 (16) | 19 (76) |
| Aztreonam-avibactam | ≤0.5/4-8/4 | 23 (92) | 2 (8) | 0 |
| Ceftazidime | >256 | 0 | 0 | 25 (100) |
| Ceftazidime-avibactam | >256/4 | 0 | – | 25 (100) |
| Ciprofloxacin | 0.5->256 | 1 (4) | 0 | 24 (96) |
| Colistin | 1–8 | 22 (88) | – | 3 (12) |
| Gentamicin | ≤0.5->256 | 5 (20) | 3 (12) | 17 (68) |
| Imipenem | 8->256 | 0 | 0 | 25 (100) |
| Meropenem | 32->256 | 0 | 0 | 25 (100) |
| Piperacillin/tazobactam | >256/4 | 0 | 0 | 25 (100) |
| Sulfamethoxazole/ trimethoprim | ≤0.5/9.5->128/2432 | 3 (12) | – | 22 (88) |
| Tigecycline | ≤0.5-1 | 25 (100) | – | 0 |

**Table 3 Pairwise SNPs between ST410 strains of this study with strain 020001 as the reference**

| | 020001 | 020026 | 020031 | 020032 | 020129 | 020147 |
|--------|--------|--------|--------|--------|--------|--------|
| 020001 | – | 293 | 292 | 226 | 299 | 286 |
| 020026 | 293 | – | 7 | 457 | 30 | 17 |
| 020031 | 292 | 7 | – | 456 | 29 | 16 |
| 020032 | 226 | 457 | 456 | – | 463 | 450 |
| 020129 | 299 | 30 | 29 | 463 | – | 21 |
| 020147 | 286 | 17 | 16 | 450 | 21 | – |

the other four. We investigated the clonal relatedness between ST410 strains in this study and other ST410 strains with genome sequences available in GenBank (Supplementary Dataset 2). From this we clearly identified the B4/H24RxC clone of ST410, which contains 37 strains including four from the present study (strains 020026, 020031, 020129, and 020147) and strains from Asia (the Philippines and Thailand), Europe (Denmark, Italy, Norway, Turkey, and UK), and North America (Canada and USA) (Fig. 1, the numbers of SNPs are shown in Supplementary Dataset 3).

**Emergence of the B4/H24RxC clone in the last decade driven by recombination.** Strain 020026, the first strain of the B4/H24RxC MDR clone in this study, was also subjected to long-read whole-genome sequencing to obtain its complete genome

sequences. A more precise phylogeny (Fig. 2) was inferred by using the complete chromosome of strain 020026 as the reference genome and using Gubbins to identify and remove recombination regions. The results showed several recombination hotspots (Supplementary Fig. 1), including genes involved in toxin–antitoxin system, flagellum, metabolism, and phage life cycle, and notably, a 13-kb region that was unique to the B4/H24RxC MDR clone. A smaller scale phylogeny re-construction was performed solely on the B4/H24RxC MDR clone, closely-related strains and the sister clade and rooted on strain E006910 (accession no. ERR1197948) of the sister clade (Supplementary Fig. 2). Recent recombination events were identified in four strains of the B4/H24RxC MDR clone. The four strains, 115102 (accession no. SRR7716572), N14-01320 (accession no. SRR5714046), EuSCAPE_SC020 (accession no. ERR1374952), and ECS1_14 (accession no. ERR2088799) had recombination regions of 94,045, 5076, 4164, and 46,701 bp, respectively. By excluding SNPs within these recent recombination regions, the average pairwise SNP distance among the 37 B4/H24RxC MDR clone strains was 31, ranging from 0 to 105 (Supplementary

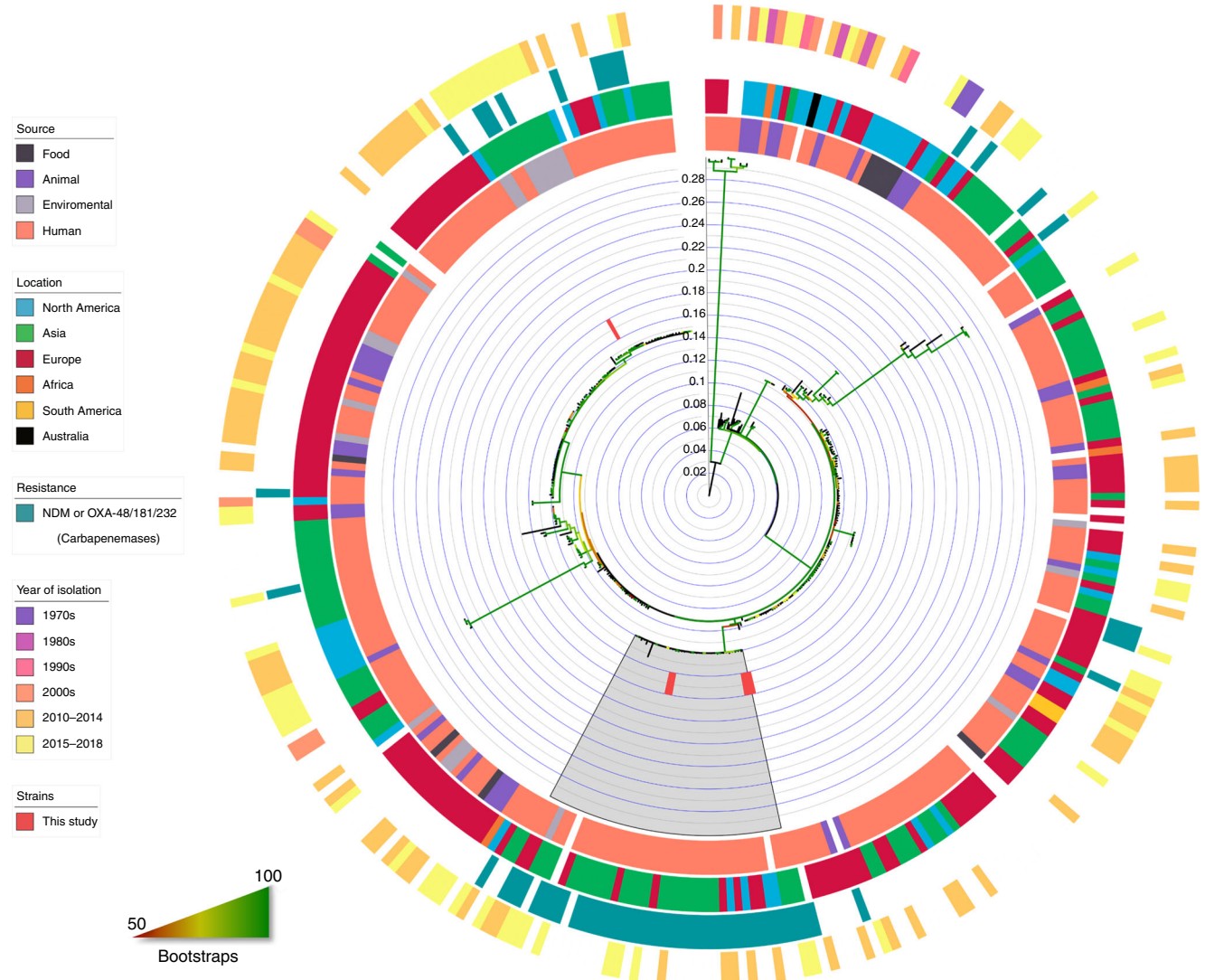

**Fig. 1** Phylogenomic tree of ST410 *E. coli* strains. A circular phylogenomic tree of ST410 *E. coli* strains ($n = 327$) was inferred using strain 020001 as the reference. Information on the strains is available in Supplementary Dataset 1 and the numbers of SNPs are shown in Supplementary Dataset 3. The six strains isolated in this study are indicated in red. The B4/H24RxC clone is also highlighted by a gray region. The four colored circles represent the source, location, resistance genes, and year of isolation from inside to outside, respectively. Bootstrap values are represented by gradient colors and a scale bar for the ST410 phylogeny is shown

Dataset 4). Given known dates and origins of isolation, such a low level of core SNPs strongly suggests recent emergence and likely on-going global dissemination of this clone. A total of 2602 SNPs were identified on the branch separating the B4/H24RxC MDR clone from its most closely-related strain (Supplementary Fig. 2), of which 2510 (96.46%) were identified to be within regions of recombination, giving a per site $r/m$ ratio (the relative likelihood that a polymorphism was introduced through recombination rather than point mutation) of 27.28. This suggests that the primary evolutionary events underpinning the emergence of the clone were driven by homologous recombination.

Coalescent analysis with all dated strains failed to converge within an applicable time (see Methods for details) during the run using the tool BactDating v1.0.1 (ref. [14]) (Supplementary Fig. 3). Four distant ST410 strains (strains KOEGE 131, MOD1-EC5419, KTE221, and NC_STEC121, see Methods for detail) were therefore excluded for dating. The refined analysis revealed an average clock rate of $\mu = 3.86$ [3.05–4.57] substitutions per year and a root date of December 1899 (95% confidence interval [95%

CI], October 1850–April 1928; Fig. 3, Point C), indicating that ST410 emerged sometime around the turn of the twentieth century. The most recent common ancestor of the B4/H24RxC MDR clone was estimated to emerge in June 2009 (95% CI, March 2007–December 2010; Fig. 3, Point A), suggesting the clone has emerged in the past 10 years.

Our data are consistent with the initial characterization of the B4/H24RxC MDR clone[9], including its recent emergence from the ST410 lineage. All but three strains had either a $bla_{NDM}$ or a $bla_{OXA-181}$-like carbapenemase ($bla_{OXA-181}$ and $bla_{OXA-232}$, two OXA-48-type carbapenemase-encoding genes; OXA-232 differs from OXA-181 by a single amino acid) or both. Although $bla_{NDM}$ and $bla_{OXA-181}$ have also been seen in strains of other lineages, half (17/34) of the $bla_{NDM}$-carrying ST410 strains sequenced, and almost all (21/23) $bla_{OXA-181}$-carrying ST410 strains belonged to the clone. Previous work by our group has demonstrated that both $bla_{NDM}$ and $bla_{OXA-181}$ are carried by IncX3 plasmids with identical backbones and swapping of the corresponding locus generates a plasmid carrying either $bla_{NDM}$ or $bla_{OXA-181}$ (ref. [8]).

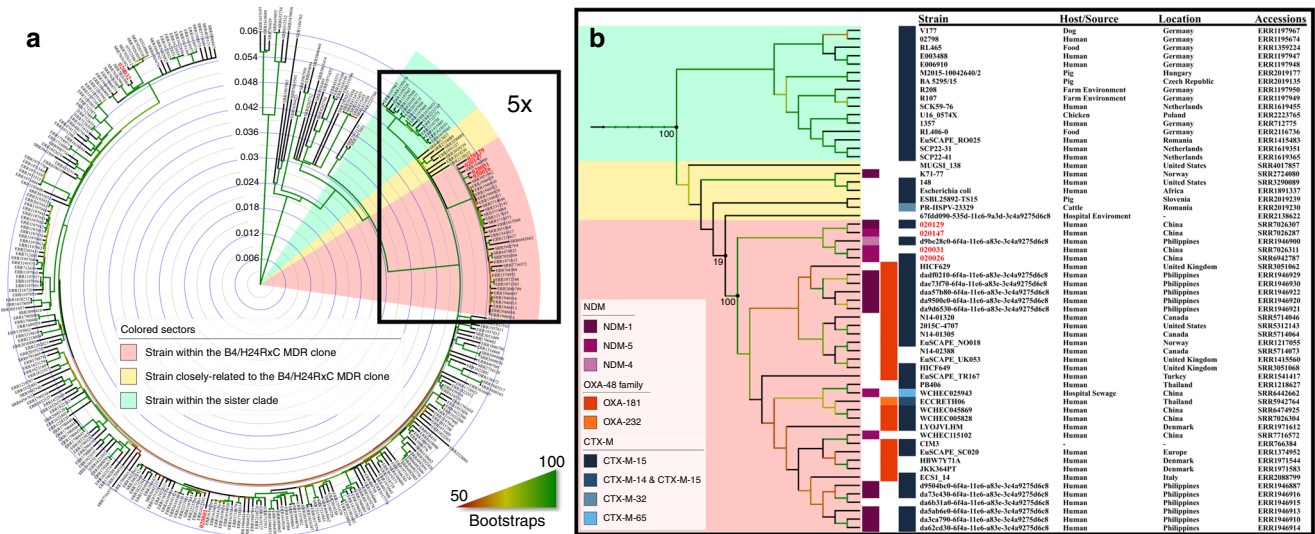

**Fig. 2** A refined Phylogenomic tree of ST410 *E. coli* strains and the emerging lineage. **a** A circular phylogenomic tree of ST410 *E. coli* strains (*n* = 327) was inferred using strain 020026 as the reference. The numbers of SNPs are shown in Supplementary Dataset 4. Several strains that were closely related to the B4/H24RxC clone are highlighted by a yellow region, while a sister clade is highlighted by a green region. Bootstrap values are represented by gradient colors and a scale bar for the ST410 phylogeny is shown. **b** A fivefold enlarged phylogenomic tree of the B4/H24RxC clone (the pink region), several closely related strains (the yellow region) and a sister clade (the green region). Strain names, sources, locations, accession numbers, carbapenemase genes, and CTX-M ESBL genes are shown

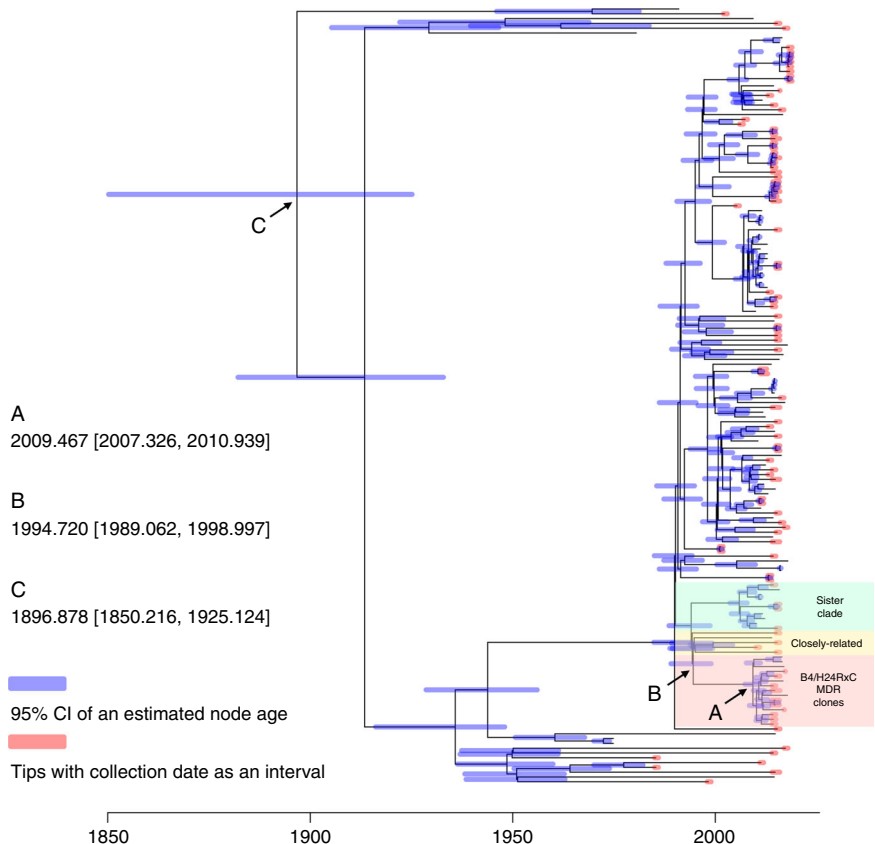

**Fig. 3** The dated phylogenomic tree of ST410 *E. coli* strains. The tree was constructed using BactDating v1.0.1 and corrected for recombination using Gubbins v2.3.4. Four strains within the B4/H24RxC MDR clone, i.e. strain KOEGE 131 (358a) (accession no. SRR785629), MOD1-EC5419 (accession no. SRR6512532), KTE221 (accession no. SRR633754,) and NC_STEC121 (accession no. SRR5470036), were excluded due to their distant relationship to the remaining clonal strains (>5000 SNPs with almost all other ST410 strains). Point A, the common ancestor of the B4/H24RxC MDR clone was estimated to emerge in June 2009 (95% CI, March 2007–December 2010). Point B, the separation of B4/H24RxC MDR clone and its closely related strains from the sister clone was estimated in September 1994 (95% CI, August 1989–December 1998) but without any other identified intermediate strain. Point C, the emergence of ST410 *E. coli* was estimated in December 1899 (95% CI, October 1850–April 1928)

In addition, most (26/37) strains within the clone also carried $bla_{CTX-M-15}$. A previous study found interspecies transmission of ST410 *E. coli* carrying a $bla_{CTX-M}$ gene between wildlife, humans, companion animals, and the environment[15]. Unlike $bla_{NDM}$ and $bla_{OXA-181}$, $bla_{CTX-M-15}$ is not largely restricted to the clone but is dispersed across the wider ST410 population.

**$bla_{NDM}$ and $bla_{OXA-181}$ are carried by IncX3 plasmids with a common backbone.** Self-transmissible plasmids carrying $bla_{NDM}$ were obtained from all six of the Sichuan province ST410 strains by conjugation and all of the plasmids had IncX3 replicons. Among the 37 strains of the B4/H24RxC clone, 17 carried $bla_{NDM}$ and 21 had $bla_{OXA-181}$ including 5 that had both $bla_{NDM}$ and $bla_{OXA-181}$. Ten of 17 $bla_{NDM}$-carrying strains had an IncX3 replicon and all 10 strains had contigs with 100% coverage and 99.93–100% identity to the reference IncX3 plasmid, pNDM5_020001, the $bla_{NDM-5}$-carrying plasmid of strain 020001 (Table 4). It is therefore likely that $bla_{NDM}$ was carried by a common pNDM5_020001-like IncX3 plasmid in the 10 strains. The complete sequence of the $bla_{NDM}$-carrying plasmid, pNDM5_020026, in strain 020026 was obtained using hybrid assembly of MinION long and Illumina short reads (Supplementary Table 5) and was indeed identical to that of pNDM5_020001. All of the 21 $bla_{OXA-181}$-carrying strains had an

IncX3 replicon. pOXA181, a $bla_{OXA-181}$-carrying plasmid, was recovered and fully sequenced from one of the 21 strains by our group as reported previously[16], and all of the remaining 20 strains had contigs with 100% coverage with pOXA181, suggesting that $bla_{OXA-181}$ was located on a common IncX3 plasmid in these strains (Table 4). pNDM5_020001 and pOXA181 have an identical IncX3 backbone with the exception of several SNPs. Therefore, it appears that a common IncX3 plasmid is frequently interchanging $bla_{OXA-181}$ and $bla_{NDM}$ genes, with both successfully co-circulating in the population. It is also possible that $bla_{OXA-181}$- and $bla_{NDM-5}$-carrying IncX3 plasmids arose independently and the plasmids are acquired interchangeably in the clone, but it is impossible to determine this from the data available.

For the five strains carrying both $bla_{NDM}$ and $bla_{OXA-181}$ with genome sequences available in GenBank, their contigs had 100% coverage with both pNDM5_020001 and pOXA181. Due to the fact that only short reads are available for these five strains, we were unable to determine whether both $bla_{NDM}$ and $bla_{OXA-181}$ were located on a single IncX3 plasmid or on different plasmids by mapping. Other approaches, such as tracking unique paths in assembly graphs from different assemblers, were attempted. However, ambiguous paths were associated with contigs containing IS26, and the largest contigs aligning with the references were not the exact size of either plasmid. Nonetheless, by comparing

### Table 4 Plasmids in strains of the lineage

| Strain | IncX3 replicon | Identity (%) with | | $bla_{NDM}$ | $bla_{OXA-181/232}$ | Other plasmid replicons |
|---|---|---|---|---|---|---|
| | | pNDM5_020001 | pOXA181 | | | |
| 020026 | + | 100 | | 5 | | Col(BS512), FIA, FIB |
| 020031 | + | 100 | | 5 | | Col(BS512), FIA, FIB |
| 020129 | + | 100 | | 1 | | Col(BS512), FIA, FIB |
| 020147 | + | 100 | | 5 | | Col(BS512), FIA, FIB |
| 025943 | + | 100 | | 5 | | Col(BS512), FII, HI2, HI2A, P1, Y |
| ERR1946920 | + | 100 | 100 | 1 | 181 | Col(BS512), C, FIA, FIB, FII |
| ERR1946921 | + | 100 | 100 | 1 | 181 | Col(BS512), C, FIA, FIB, FII |
| ERR1946922 | + | 100 | 100 | 1 | 181 | Col(BS512), C, FIA, FIB, FII |
| ERR1946929 | + | 100 | 100 | 1 | 181 | Col(BS512), C, FIA, FIB, FII, Q1 |
| ERR1946930 | + | 100 | 100 | 1 | 181 | Col(BS512), C, FIA, FIB, FII, Q1 |
| 045869 | + | | 100 | | 181 | Col(BS512), FIA, FIB |
| 005828 | + | | 100 | | 181 | Col(BS512), FIA, FIB |
| ERR1217055 | + | | 100 | | 181 | Col(BS512), FIA, FIB |
| ERR1415560 | + | | 100 | | 181 | Col(BS512), FIA, FIB, FII |
| ERR1541417 | + | | 100 | | 181 | FIA, FII, L/M |
| ERR1971544 | + | | 100 | | 181 | Col(BS512), FIB |
| ERR1971583 | + | | 100 | | 181 | Col(BS512), FIA, FIB, FII |
| ERR1971612 | + | | 100 | | 181 | Col(BS512), FIB |
| ERR2088799 | + | | 100 | | 181 | Col(BS512), FIA, FIB, FII |
| ERR766384 | + | | 100 | | 181 | Col(BS512), FIA, FIB, FII |
| SRR3051062 | + | | 100 | | 181 | Col(BS512), FIA, FIB, FII |
| SRR3051068 | + | | 100 | | 181 | Col(BS512), FIA, FIB, FII |
| SRR5312143 | + | | 100 | | 181 | Col(BS512), Col156, FIA, FIB, FII |
| SRR5714046 | + | | 100 | | 181 | Col(BS512), FIA, FIB, FII |
| SRR5714064 | + | | 100 | | 181 | Col(BS512), Col(IMGS31), FIA, FIB, FII |
| SRR5714073 | + | | 100 | | 181 | Col(BS512), FIA, FIB, FII, I1, Y |
| 115102 | | | | 5 | | Col(BS512), FIA, FIB |
| ERR1946887 | | | | 1 | | Col(BS512), C, FIA, FIB, FII |
| ERR1946900 | | | | 4 | | Col(BS512), FIA, FIB, FII, Y |
| ERR1946910 | | | | 1 | | Col(BS512), C, FIA, FIB, FII |
| ERR1946913 | | | | 1 | | Col(BS512), C, FIA, FIB, FII |
| ERR1946914 | | | | 1 | | Col(BS512), C, FIA, FIB, FII |
| ERR1946915 | | | | | | Col(BS512), FIA, FIB, FII, X4 |
| ERR1946916 | | | | 1 | | Col(BS512), C, FIA, FIB, FII, X4 |
| ERR1218627 | | | | | | None |
| ERR1374952 | + | | | | | Col(BS512), FIA, FIB, FII |
| SRR5942764 | | | | | 232 | Col(BS512), ColKP3, ColpVC |

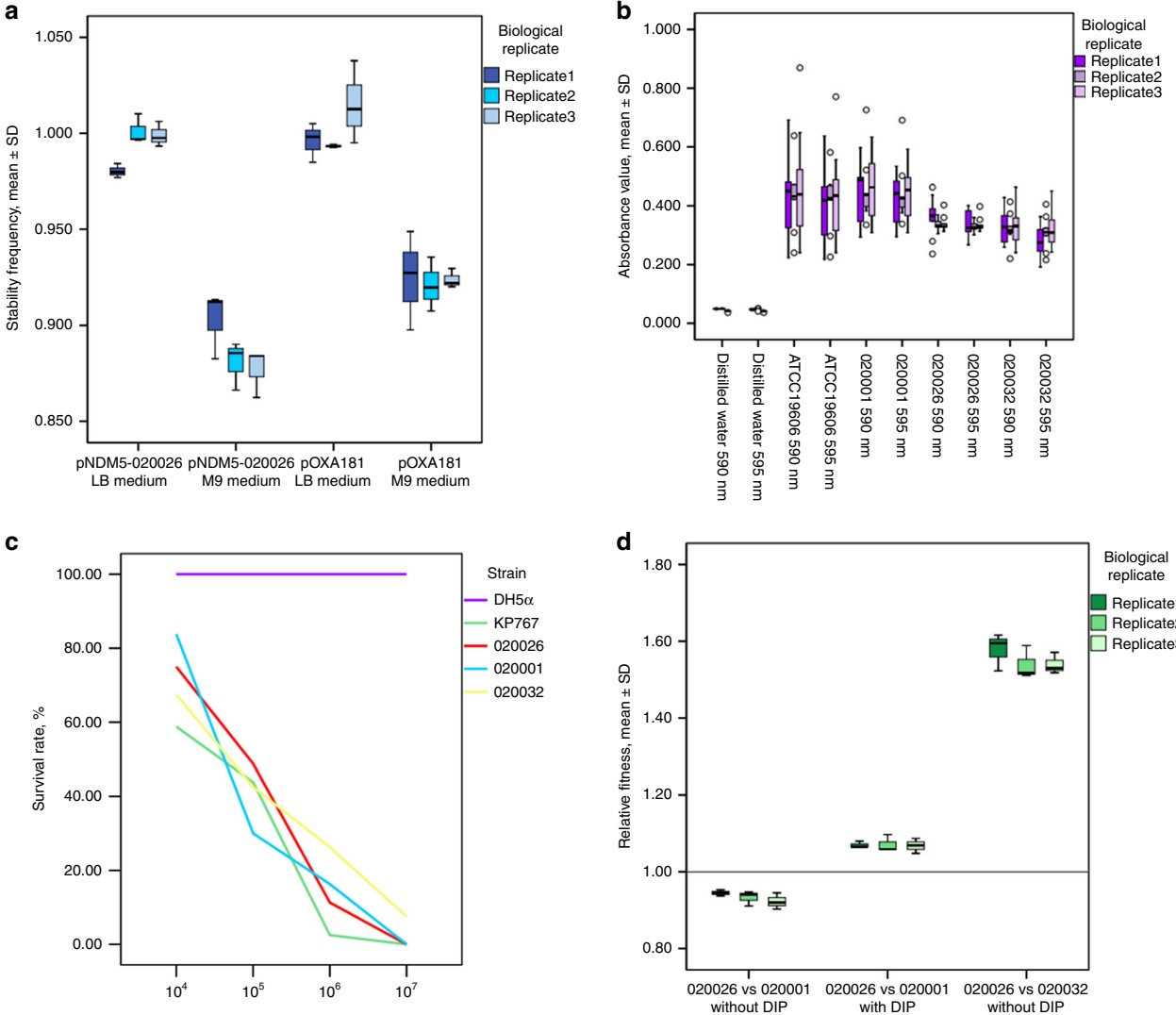

**Fig. 4** Plasmid stability, biofilm formation, virulence assay, and strain relative fitness. **a** Stability of pNDM5_020026 and pOXA181 in *E. coli* J53. The mean ± standard deviation (SD) of the stability frequency is shown. $N = 3$ biologically independent experiments. The results are also shown in Supplementary Table 7. **b** Biofilm formation of bacterial strains. Absorption values of strains 020001, 020026, and 020032 at both OD$_{590\,nm}$ and OD$_{595\,nm}$ are shown. Strain ATCC 19606 and H$_2$O were used as the positive and negative control, respectively. The results are also shown in Supplementary Table 8. $N = 3$ biologically independent experiments. **c** Survival of *G. mellonella* after infection by bacterial strains. The effect of $1 \times 10^4$, $1 \times 10^5$, $1 \times 10^6$, and $1 \times 10^7$ CFU of each strain on survival of *G. mellonella* at 72 h after infection is shown. The exact survival rates are also shown in Supplementary Table 8. KP767, a hypervirulent *K. pneumoniae*, was used as a positive control, while *E. coli* DH5α was used as a negative control. $N = 5$ biologically independent experiments. The results are also shown in Supplementary Table 9. **d** Relative fitness of strain 020026 compared to strain 020001 and strain 020032. The competition between strain 020026 and strain 020001 was also performed in the presence of 375 DIP to create iron-deprived conditions. $N = 3$ biologically independent experiments. The results are also shown in Supplementary Table 10

and contrasting the predicted plasmid replicons of the strains within the same clade, it could be deduced that *bla*$_{NDM-1}$ was carried on an IncA/C plasmid, while *bla*$_{OXA-181}$ was carried on an IncX3 plasmid, in the five strains carrying both *bla*$_{NDM-1}$ and *bla*$_{OXA-181}$. Seven *bla*$_{NDM}$-carrying strains within the clone did not have an IncX3 replicon, suggesting that *bla*$_{NDM}$ was carried by plasmids of other replicon types and strains of the clone have acquired *bla*$_{NDM}$ more than once.

**IncX3 plasmids were stably maintained in nutrient-rich media.** We performed plasmid stability tests for representative IncX3 plasmids carrying *bla*$_{NDM-5}$ or *bla*$_{OXA-181}$ from strains of the B4/H24RxC MDR clone. In LB media (representing nutrient-rich settings), the stability frequency of pNDM5_020026 and pOXA-181 was 0.99 ± 0.01 and 1.00 ± 0.01 (mean ± standard derivation

[SD]; Fig. 4a and Supplementary Table 7), respectively. The plasmid loss rate of pNDM5_020026 and pOXA-181 in LB media was 1.22% ± 0.77% and 1.11% ± 0.51%, respectively. The stability frequency in M9 minimal media (representing nutrient-restricted settings) of pNDM5_020026 and pOXA-181 was 0.89 ± 0.01 and 0.92 ± 0.00 and the plasmid loss rate of the two plasmids was 27.33 ± 3.18% and 24.11 ± 1.71%, respectively. The above findings suggest that IncX3 plasmids carrying *bla*$_{NDM-5}$ or *bla*$_{OXA-181}$ were stably maintained at equal frequencies in nutrient-rich settings. Our data also suggest that these plasmids were prone to loss at an elevated frequency in nutrient-restricted settings.

**B4/H24RxC-specific genes encoding adherence and iron acquisition.** All strains of the B4/H24RxC clone had three genes that had no orthologous genes with >90% nucleotide identity

present in all other ST410 strains. The first gene is *yadC*, which encodes a fimbriae-like protein YadC (NCBI Reference Sequence accession no. WP_000848455.1). *yadC* was absent from strain 020001, while strain 020032, which is a ST410 CREC identified in the present study but is more phylogenetically distinct from B4/H24RxC than strain 020001 (Fig. 2), had a gene with 76% coverage and 56.5% identity. YadC has been purported to be involved in adhesion, internalization, and motility of *E. coli* and contribute to its pathogenicity[17]. The second gene is *ybjI*, which encodes a pentapeptide repeat-containing protein (NCBI Reference Sequence accession no. WP_000868898.1) but could be a pseudogene (https://www.uniprot.org/uniprot/P32690). *ybjI* was absent from both strain 020001 and strain 020032. The third gene, *fhuA*, encodes a ferrichrome porin FhuA (NCBI Reference Sequence accession no. WP_039023099.1) and constitutes the *fhu* operon with *fhuB*, *fhuC*, and *fhuD*. The *fhu* operon is essential for the utilization of ferric siderophores of the hydroxamate type[18] and also contributes to bacterial virulence[19]. There are multiple types of FhuA-like proteins in *E. coli*. The remaining 290 ST410 strains including strain 020001 and strain 020032 contained a gene encoding an alternative allele of FhuA (NCBI Reference Sequence accession no. WP_000124383.1; Supplementary Fig. 4), which had <74% nucleotide identity with the

clone-specific *fhuA*. Blast analysis of the clone-specific *fhu* operon showed a 100% nucleotide identity match with the *fhu* operon of numerous other *E. coli* strains, suggesting that the allelic replacement of the *fhu* operon was derived from recent recombination in the B4/H24RxC MDR clone. The three gene alleles (*yadC*, *ybjI*, and *fhuA*) were present in all strains of the B4/H24RxC MDR clone while absent from all other ST410 strains, suggesting that the alleles were acquired at the emergence of the B4/H24RxC MDR clone and then swept to fixation in the clone.

**B4/H24RxC-specific SNPs associated with adherence and iron acquisition.** We further identified SNPs unique to the B4/H24RxC clone. A total of 382 SNPs were present in all members of the clone but absent from other ST410 genomes (Supplementary Dataset 5). Among the 382 SNPs, 362 were in coding sequences but only 60 were non-synonymous substitutions, present in 48 genes (Fig. 5 and Supplementary Dataset 5). Most of the 48 genes encode products involved in metabolism including three dehydrogenases involved in anaerobic metabolism. In addition, we found 20 clone-specific SNPs in intergenic regions, which have been shown to be under strong evolutionary constraints[20]. Nine clone-specific SNPs in intergenic regions were

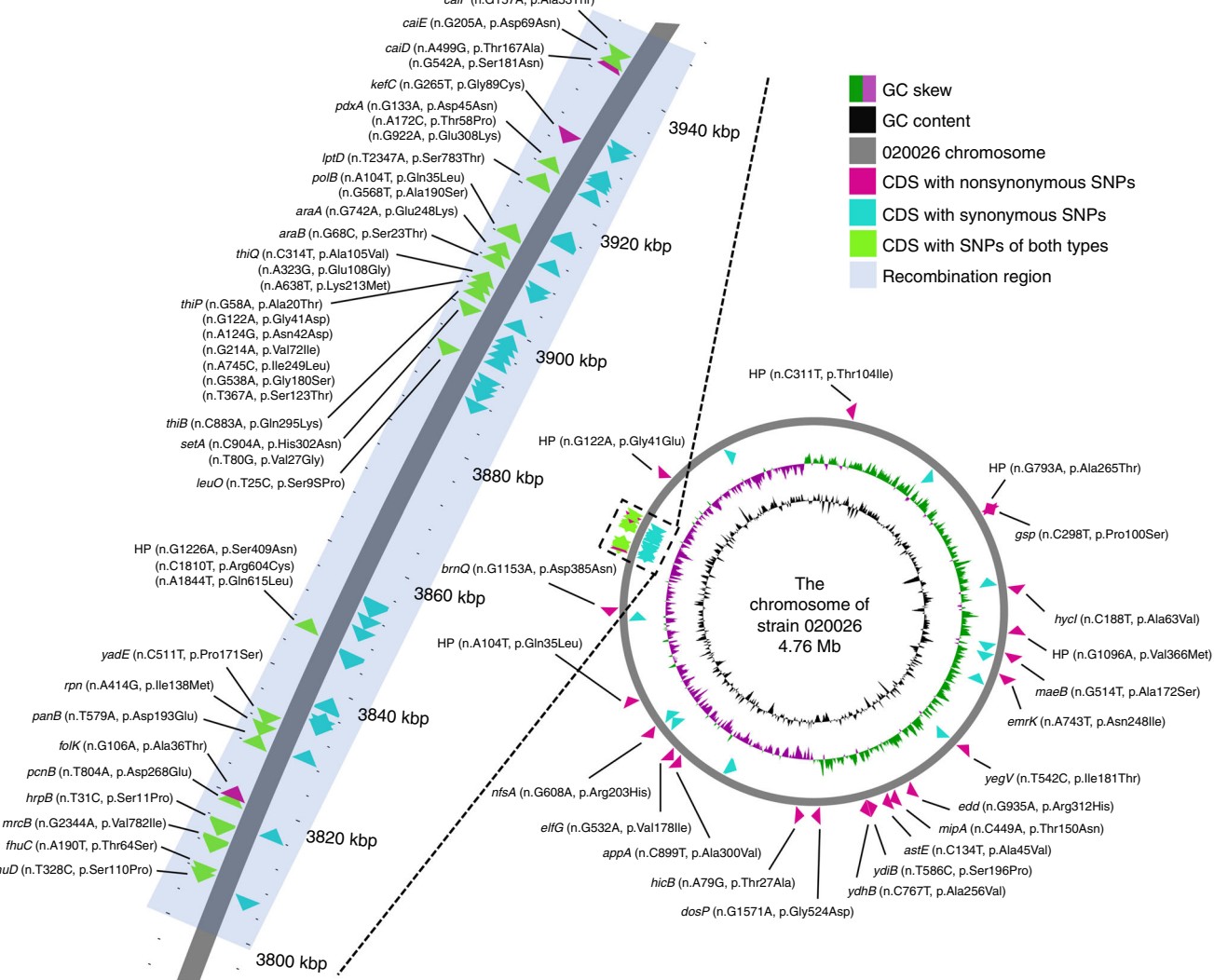

**Fig. 5** Location of all synonymous and non-synonymous SNPs unique to the emerging MDR clone compared with other ST410 strains. The circular chromosome diagram was generated using CGView[62]. The detailed information of the SNPs is available in Supplementary Dataset 5

**Table 5 Iron source growth assay results in the presence of DIP at the MIC (500 μM for strains 02001 and 020026, and 250 μM for strain 020032)**

| Strain | Bovine serum albumin (10 mg/ml) | FeCl₂ (1 mM) | Hemin (10 μM) | Hemoglobin (1 mg/ml) | Holo-transferrin (10 mg/ml) | Lactoferrin (10 mg/ml) |
|---|---|---|---|---|---|---|
| 020001 | − | + | − | + | − | + |
| 020026 | + | + | + | − | − | + |
| 020032 | − | + | − | − | − | − |

located in the -10, -35 boxes of promoter such as those found in the upstream of ferrichrome porin gene *fhuA*, recombination-promoting family gene *rpn*, and aspartate decarboxylase gene *panD*, or within the 5′ UTR of downstream genes such as glucose uptake transporter regulator *sgrR* and inhibitor *sgrT* (Supplementary Dataset 5). Previously, we have also found unique intergenic SNPs and unique gene alleles encoding anaerobic metabolism in ST167 and ST617 (ref. [8]), as well as showing that these are key evolutionary events in the emergence of the globally disseminated ST131 clone C[21]. Several genes encoding clone-specific SNPs (*elfG, fhuC, fhuD, lptD,* and *emrK*) may promote the survival of bacteria. The *elfG* gene is part of the *elfADCG-ycbUVF* fimbrial operon-encoding proteins which promote adhesion of bacterial cells to abiotic surfaces[22] and may therefore facilitate bacteria to colonize the wider hospital environment. *fhuC* and *fhuD* are part of the *fhu* operon[19] as described above. *lptD* (also known as *imp*) encodes the lipopolysaccharide-assembly protein LptD, which is essential for integrity of the membrane and is related to the sensitivity of bacteria to detergents, antibiotics, and dyes[23,24]. *emrK* encodes part of a tripartite efflux system named EmrYK-TolC, which confers stress-inducible functions including those imposed by antimicrobial agents to reduce the lethal effects[25,26]. Further analysis of the genomic location of these clone defining SNPs identified clear clustering of the SNPs in a defined region of the chromosome. Confirmatory analysis of the recombination detection tests using Gubbins[27] on the reference-mapping based-SNP alignment demonstrated that over 90% ($n = 346$) clone-specific SNPs, 65% ($n = 39$) clone-specific non-synonymous SNPs had been introduced via a recombination event or events, facilitating a more rapid adaptation and evolutionary emergence of the MDR clone (Supplementary Fig. 1).

**Decreased ability of the B4/H24RxC clone to form biofilms.** Given the unique *yadC* gene and unique SNPs in *elfG* we performed assays of biofilm formation to plastic abiotic surfaces for a representative strain (020026) of the clone with two non-clone ST410 strains, 020001 and 020032, as control. Strain 020026 exhibited significantly less biofilm formation (e.g. absorption at $OD_{590\,nm}$, mean ± standard deviations, $0.35 ± 0.05$ vs $0.46 ± 0.11$, $P < 0.001$; Fig. 4b and Supplementary Table 8) than strain 020001. Therefore, the unique gene and SNPs seen in the clone significantly decrease adherence to abiotic surfaces and the ability to form a biofilm. Adherence is a key factor for bacteria to colonize hosts including humans[28]. Although the clone has a decreased ability to form biofilms on abiotic surfaces, it still has the ability to form biofilms and does not lose adherence, meaning that it is still capable of colonizing hosts.

**The B4/H24RxC clone exhibits enhanced ability to utilize iron sources.** Given our identification of clone-specific SNPs in the *fhu* operon, we performed an iron source growth assay and found that the presence of 250 μM DIP completely inhibited growth of strain 020032. By contrast, the MIC of 2′2-dipyridyl (DIP) for strain 020001 and 020026 was 500 μM DIP, suggesting that

the two strains were more resistant to iron-deprived conditions than strain 020032. The addition of FeCl₂ restored growth of all three strains, while the addition of lactoferrin only restored growth of strain 020001 and strain 020026 but not that of strain 020032 (Table 5). Lactoferrin is present on mucous membranes and is part of the human innate defense[29]. The ability to utilize iron from lactoferrin may therefore facilitate bacteria to colonize the human gut. The enhanced resistance to iron-deprived conditions and the ability to sequester iron from lactoferrin allow strain 020001 and strain 020026 to adapt to the human host better than the more distant strain 020032. The addition of hemin and bovine serum albumin restored growth of strain 020026 but not strain 020001 nor strain 020032, while the addition of hemoglobin only restored growth of strain 020001 (Table 5). Hemin (ferric chloride heme) is an oxidized form of heme and is produced during processing aged red blood cells[30]. Serum albumin is the most abundant blood protein in humans. The utilization of hemin and serum albumin as the sole iron source may therefore facilitate the survival of B4/H24RxC strains in human hosts.

**The B4/H24RxC clone does not have enhanced virulence.** The 50% lethal dose ($LD_{50}$) at 72 h of strains 020026, 020001, and 020032 against *G. mellonella* were identical at $1 × 10^5$ CFU, also identical to that of the hypervirulent *K. pneumoniae* strain KP767 (Fig. 4c and Supplementary Table 9). Therefore, the B4/H24RxC strain 020026 displays virulence comparable to other members of the ST410 lineage, but no obvious enhancement of the virulence phenotype. It is well known that the *fhu* iron acquisition operon contributes to bacterial virulence[19,31]. However, the clone-specific *fhuA* and the clone-unique SNPs in *fhuC* and *fhuD* of the emerging MDR clone do not lead to enhanced virulence.

**The B4/H24RxC clone shows a fitness advantage in iron-deprived conditions.** Strain 020026 exhibited a fitness advantage compared to strain 020032 (relative fitness value [$w$], $1.28 ± 0.06$; mean ± standard deviation; Fig. 4d and Supplementary Table 10) but was slightly outcompeted by strain 020001 ($w$, $0.93 ± 0.02$; Fig. 4 and Supplementary Table 10) in LB media. As strain 020026 was able to utilize more iron sources than 020001, we also performed competition experiments between strain 020026 and 020001 under iron-deprived conditions. Strain 020026 outcompeted strain 020001 ($w$, $1.07 ± 0.02$; Fig. 4d and Supplementary Table 10) in iron-deprived media. The difference of the $w$ values of strain 020026 compared to strain 020001 between iron-deprived and non-iron-deprived conditions was statistically significant ($t = 17.33$, $P < 0.001$). The fitness advantage in iron-deprived environments seen in strain 020026 is therefore likely to promote the survival and persistence of B4/H24RxC strains in human hosts.

**Discussion**
Our data presented here stemmed from a genomic epidemiology and surveillance study of CREC in Sichuan Province, China. The identification of ST410, ST167, and ST617 as dominant CREC

clones in the province led us to comprehensively characterize the ST410 lineage (having previously characterized the ST167 and ST617 lineages)[8]. Phylogenomics revealed that the majority of Chinese CREC ST410 belonged to a previously identified, globally disseminated clone of CREC ST410 labeled B4/H24RxC[9]. By using MinION sequencing in combination with the available Illumina genome data, we were able to additionally show that the clone is dominated by an IncX3 plasmid which has expanded with the clone, but which frequently interchanges the carbapenemase genes $bla_{NDM-5}$ and $bla_{OXA-181}$ without any impact on plasmid stability or fitness. In an effort to identify key evolutionary events in the emergence of the B4/H24RxC clone, we uncovered a number of SNPs and core-gene alleles unique to the clone in comparison to the remainder of the ST410 lineage. Our findings show unique SNPs in core anaerobic metabolism genes and intergenic regions within the B4/H24RxC clone. These have been shown to be key events in the emergence of MDR ST167 and ST617 lineages, as well as the MDR ST131 clone C[21]. Therefore, our data add further compelling evidence to the notion that evolution of MDR in *E. coli* is parallel in nature and as such predictable. In this study we also show for the first time how SNPs and gene alleles associated with increased colonization of mammalian hosts are associated with fundamental changes in important phenotypes. Indeed, our data are indicative of a scenario where emerging clones of MDR *E. coli* accumulate SNPs enhancing key phenotypes such as iron acquisition, while abrogating phenotypes are more associated with environmental survival such as adhesion to abiotic surfaces. This concept of adaptation to the human host is further supported by the fact that in laboratory media competition experiments, the MDR B4/H24RxC clone of ST410 is slightly outcompeted by strains of other clones within ST410 but has advantage under iron-deprived conditions, suggesting that the fitness associated with SNPs affecting colonization potential are advantageous within the human clinical environment. These findings present a vitally important new direction in our understanding of the emergence and dynamics of clones of MDR *E. coli*.

## Materials and methods

**Strain isolation and in vitro susceptibility testing**. All non-duplicate CREC clinical strains ($n = 25$) were collected from one referral and seven municipal hospitals in Sichuan Province, China, between June 2016 and February 2017 (information about the hospitals is available in Supplementary Table 1). This study was approved by the ethical committee of West China Hospital and informed consents were waived. All of the strains were initially identified as *E. coli* using Vitek II (bioMérieux; Marcy-l'Étoile, France). The strains were isolated from various clinical samples including blood ($n = 8$), sputum ($n = 6$), and urine ($n = 5$) (Table 1 and Supplementary Dataset 1). MICs of antimicrobial agents were determined using the microdilution method of the Clinical and Laboratory Standards Institute (CLSI)[32]. For ceftazidime/avibactam, colistin, and tigecycline, the breakpoints defined by the European Committee on Antimicrobial Susceptibility Testing (EUCAST) were used, while the breakpoints of aztreonam were applied for aztreonam–avibactam.

**Whole-genome sequencing and analysis**. All Chinese strains isolated in this study ($n = 25$) were subjected to whole-genome sequencing using the HiSeq X10 (Illumina; San Diego, CA, USA) according to the manufacturer's instructions. Genomic DNA was prepared using the QIAamp DNA Mini Kit (Qiagen, Hilden, Germany). Generated reads were subjected to strict quality-control filtering including trimming 10 bases from each end and bases with quality below Q15, removing adaptor sequences and discarding reads with an average quality below Q20 using Cutadapt v1.16 (ref. [33]) and BBTools v37.92. The reads were then de novo assembled into contigs using SPAdes v3.13.0 (ref. [34]) applying the careful and auto-cutoff modes. Strain 020001 (the first ST410 strain recovered in this study), 020026 (the first strain of the B4/H24RxC MDR clone identified in this study, see below), and 020032 (a phylogenetically distant strain within ST410 isolated in this study) were also subjected to whole-genome sequencing using the long-read MinION Sequencer (Nanopore; Oxford, UK). Libraries were constructed using the SQK-LSK109 kit and were multiplexed using native barcodes from the EXP-NBD104 kit, according to protocols of the manufacturer (Nanopore). Sequencing was performed in an R9.4.1 Flow Cell for 48 h and the yield of MinION sequencing

data of the three strains is shown in Supplementary Table 4. The MinION reads were base-called and demultiplexed using Guppy v3.0.3. Reads with adaptors at the ends were trimmed and those with adapters in the middle were discarded using Porechop v0.2.4. A de novo hybrid assembly of both short Illumina reads and long MinION reads was performed using Unicycler v0.4.7 (ref. [35]) under conservative mode for increased accuracy. Complete circular contigs were then corrected and polished using Pilon v1.22 (ref. [36]), in addition to the integrated polishing steps in Unicycler, with Illumina reads for several rounds (7 on average) until no further improvements were reported. Prokka v1.13 (ref. [37]) was used to annotate the genome sequence. Acquired antimicrobial resistance genes were identified using ResFinder v3.1 (http://genomicepidemiology.org/). STs were assigned by querying the multi-locus sequence typing database of *E. coli* (http://enterobase.warwick.ac.uk/species/index/ecoli).

**Determining clonal relatedness by SNPs analysis**. The chromosomal sequences of strain 020001 and 020026 obtained from the hybrid assembly of MinION/Illumina sequencing reads were used as the reference for mapping. Reads passing the quality-control thresholds aligned to the reference using Snippy v4.3.6 with default settings. Aligned pseudo-genomes were created and cleaned using the integrated scripts provided by Snippy v4.3.6. Phage regions (Supplementary Table 6) were identified using the PHASTER server[38] and intact phage regions were masked using "N" with other settings as default. Recombination regions and a recombination-corrected phylogenetic tree were identified and inferred using Gubbins v2.3.4 (ref. [27]) with the GTRGAMMA model and a maximum of 50 iterations. Matrix representing pairwise SNP distance was calculated using snp-dists v0.6.3.

**Determining the population structure of ST410**. All ST410 genome sequences with short reads available in GenBank ($n = 327$; Supplementary Dataset 2, accessed 1 August 2018) were retrieved from either the EnteroBase collection or from NCBI SRA database. Strict quality-control, de novo assembly and SNP calling were also performed on these reads as described above. Six strains (Supplementary Dataset 2) had >15% undetermined sites, which are shown as "N" in their genome sequences, in the pseudo-genome and were therefore excluded from all further analyses. A precise phylogeny was obtained by masking SNP sites residing in recombination regions using Gubbins v2.3.4 (ref. [27]) as described above and the output phylogenetic tree was tested using bootstrapping ($n = 1000$) in RAxML v8.2.12 (ref. [39]) under the GTRGAMMA model. The phylogenetic tree of ST410 genomes was visualized and annotated using iTOL v3 (ref. [40]) and Phandango v1.3.0 (ref. [41]).

**Coalescent analysis of dated ST410 strains**. Dated strains (Supplementary Dataset 2) with either a specific or an interval of time in unit of years were fed into Gubbins v2.3.4 (ref. [27]) as described above to obtain a recombination-corrected tree, which was then used as the input in BactDating v1.0.1 (ref. [14]) under mixed model with $10^8$ iterations to ensure that the Markov chain Monte Carlo (MCMC) was run for long enough to converge (the effective sample size of the inferred parameters $\alpha$, $\mu$, and $\sigma$ were >200). In addition, a coalescent analysis without the most distant ST410 strains including KOEGE 131 (358a) (accession no. SRR785629), MOD1-EC5419 (accession no. SRR6512532), KTE221 (accession no. SRR633754), and NC_STEC121 (accession no. SRR5470036) to accelerate convergence was performed using the same settings. The four strains do not belong to the B4/H24RxC MDR clone and are >5000 SNPs distant from all other ST410 strains (Supplementary Dataset 4).

**Identifying loci specific to the B4/H24RxC MDR clone**. The genome sequences of all ST410 strains ($n = 327$) were annotated using Prokka v1.13 (ref. [37]). A pan-genome matrix of the 327 ST410 genomes was obtained using Roary v3.12.0 (ref. [42]) with 90% used as the minimum amino acid identity, followed by a clone-association analysis using Scoary v1.6.16 (ref. [43]) with a maximum adjusted p-value of 1e-30. *fimH* typing of ST410 strains was performed with FimTyper 1.0 (ref. [44]). Genes present in the B4/H24RxC MDR clone ($n = 37$) but absent from all other ST410 genomes ($n = 290$), and vice versa, were considered as clone-specific. SNPs specific to the B4/H24RxC MDR clone were identified by feeding the entire SNP matrix of all ST410 strains into Scoary v1.6.16 (ref. [43]) using the same settings as described above. SNPs over-represented in strains of the clone but absent from all other ST410 strains were considered as specific for the B4/H24RxC MDR clone. Gubbins v2.3.4 (ref. [27]) was used to determine whether the clone-specific genes and SNPs were due to recombination. SNPs in the -10, -35 boxes of promoter or in the 5′ UTR regions of downstream genes were predicted using the online tool BPROM (http://www.softberry.com/)[45].

**Analysis of IncX3 plasmids within the B4/H24RxC MDR clone**. All strains within the B4/H24RxC MDR clone of ST410 ($n = 37$) were screened for plasmid replicons using PlasmidFinder v2.0 and antimicrobial resistance genes identified using ResFinder v3.1. For strains containing an IncX3 replicon and $bla_{NDM}$ or $bla_{OXA-181}$, their assembly graphs generated from SPAdes v3.13.0 and Unicycler v0.4.7 were investigated using Bandage v0.8.1 (ref. [46]) by tracking paths flanking the resistance gene-containing and the replicon-containing contigs. Filtered reads of these strains were mapped against either plasmid pNDM5_020001 (accession

no. CP032424; for $bla_{NDM}$-carrying strains) or plasmid pOXA181 (accession no. KP400525; for $bla_{OXA-181}$-carrying strains)[16] or both plasmids (for strains carrying both $bla_{NDM}$ and $bla_{OXA-181}$ using BWA v0.7.17 (ref. [47]) with default settings. pNDM5_020001 is a $bla_{NDM}$-carrying IncX3 plasmid of strain 020001, while plasmid pOXA181 (accession no. KP400525) is a $bla_{OXA-181}$-carrying IncX3 plasmid of strain 005828 (ref. [16]), both of which are within the B4/H24RxC MDR clone. Contigs representing part of the complete plasmids were retrieved from entire genomes using standalone nucleotide BLAST v2.7.1 (ref. [48]).

**Plasmid mobility testing.** Conjugation experiments were carried out in broth and on filters with the azide-resistant *E. coli* strain J53 as the recipient as described previously[49–51]. For filter-based mating, overnight donor cultures (1 ml) were harvested by centrifugation, washed twice with 1 ml saline and re-suspended in 100 μl saline. Recipient cells were harvested from plates using a bent Pasteur pipette, washed, and suspended in 500 μl saline. Donor and recipient suspensions were mixed (50 μl each). The mixture was placed on a 0.45 μM cellulose-ester filter (Xinya; Shanghai, China) and then incubated on a blood agar plate at 37 °C for 4 h. Subsequently, the mixture of cultures was harvested in 1 ml saline, centrifuged, and re-suspended in 200 μl saline. For broth-based mating, overnight cultures of donor (25 μl) and recipient strains (250 μl) were added to 3 ml fresh BHI broth. The mixture was incubated for 18 h at 37 °C without shaking. Potential transconjugants were selected on LB agar plates containing 4 μg/ml meropenem and 150 μg/ml azide. The presence of $bla_{NDM}$ and the IncX3 replicon was verified by PCR with primers NDM-up (5′-TCGCCCCATATTTTTGCTAC)/NDM-dw (5′-CTGGGT CGAGGTCAGGATAG) for $bla_{NDM}$[52] and self-designed primers IncX3_forward (5′-GTTTTCTCCACGCCCTTGTTCA)/IncX3_reverse (5′-CTTTGTGCTTGG CTATCATAA) for the IncX3 replicon.

**Plasmid stability testing.** Plasmid pNDM5_020026, which was the $bla_{NDM-5}$-carrying self-transmissible plasmid from strain 020026, and $bla_{OXA-181}$-carrying plasmid pOXA-181 from strain 005828[16] were selected as representatives for testing plasmid stability. Both pNDM5_020026 and pOXA-181 are IncX3 replicon type plasmids. Plasmid stability was tested in LB broth and minimal media (M9 with 0.2% glucose, which is simply referred as M9 broth in this study) to reflect both nutrient-rich and nutrient-restricted settings as described previously[53]. Briefly, *E. coli* J53 containing pNDM5_020026 or pOXA-181 was inoculated at 37 °C overnight in 15 ml LB or M9 broth in the presence of 150 μg/ml sodium azide plus 2 μg/ml meropenem (for strain with pNDM5_020026) or plus 0.5 μg/ml imipenem (for strain with pOXA-181). These cultures were washed with saline (0.9% NaCl) several times to remove carbapenems and sodium azide. Aliquots (15 μl) were added to 15 ml LB or M9 broth correspondingly. After incubation at 37 °C for 24 h with shaking at 200 r.p.m., 100 μl samples were collected, diluted 1:10⁴ with LB or M9 broth after being measured to the 0.5 McFarland standard, and were then streaked onto LB or M9 agar plates with and without 2 μg/ml meropenem (for strain with pNDM5_020026) or 0.5 μg/ml imipenem (for strain with pOXA-181). The stability frequency of plasmids was calculated by $\log_{10}(Ng)/\log_{10}(Nw)$, where $Ng$ and $Nw$ represent number of bacterial cells containing the plasmid and all bacterial cells in the media, respectively. In addition, 100 colonies from the agar plate without meropenem were randomly selected and streaked onto plates containing 2 μg/ml meropenem to calculate the percentage of plasmid loss. For each of the two plasmids in either LB or M9 medium, 300 colonies were selected in total from triplicate repeats of each experiment.

**Biofilm formation assays.** Strain 020026 was subjected to a biofilm formation assay with strains 020001 and 020032 used as controls as described previously[54,55]. Briefly, bacterial cells were harvested from overnight cultures in LB broth by centrifugation at 2500 r.p.m. for 10 min, were re-suspended with saline, and were adjusted to 0.5 McFarland standard. Aliquots (100 μl) were then pipetted into 96-well polystyrene culture plates and were incubated for 3 h at 37 °C to allow the formation of biofilms. The plates were washed twice with distilled water. Biofilms in the wells were fixed with 100 μl methanol per well for 15 min and were stained with 100 μl staining buffer containing 1% crystal violet for 5 min. The stained biofilms were washed again to remove the unbound stain and allowed to dry at room temperature. Biofilms were detected with 110 μl 33% glacial acetic acid by ELX800 Universal Microplate Reader (Bio-Tek, Winooski, VT, USA) at OD595 nm and OD595 nm. *Acinetobacter baumannii* strain ATCC 19606 was used as a positive control, while distilled water was used as the negative control. Absorption values of strain 020026 were compared with those of strain 020001 and 020032 separately using analysis of variance with the least significant difference method.

**Virulence assay.** Wax moth (*Galleria mellonella*) larvae weighing 250–350 mg (Tianjin Huiyude Biotech Company, Tianjin, China) were used to assess the virulence of strains 020026, 020001, and 020032. A hypervirulent *K. pneumoniae* strain, KP767 (ref. [56]), was used as a positive control, while *E. coli* DH5α was used as a negative control. Overnight bacterial cultures were washed using phosphate-buffered saline (PBS) and were further adjusted with PBS to concentrations of $1 \times 10^6$ CFU/ml, $1 \times 10^7$ CFU/ml, $1 \times 10^8$ CFU/ml, and $1 \times 10^9$ CFU/ml. Larvae

($n = 16$) were injected with 10 μl of inoculum into hemocoel via the last left proleg using a 25-μl Hamilton syringe[57]. The infected larvae were then incubated in plastic containers at 37 °C. The number of live larvae was counted every 12 h for 3 days.

**Iron source growth assays.** The growth of strains 020026, 020001, and 020032 in the presence of 200, 250, 375, and 500 μM 2′2-dipyridyl (DIP; Sigma, St. Louis, MO, USA) in LB agar plates was examined to determine the MIC of DIP. Growth assays of strain 020026 under different iron sources were performed as described previously[58] with strain 020001 and strain 020032 used as controls. Briefly, prior to inoculation, bacterial strains were cultured in LB broth containing 200 μM DIP, which was lower than MIC, for 6 h to limit growth of the strains and were then washed in PBS. Approximately $10^5$ CFU of each strain were streaked onto LB agar plates in the presence of DIP at the MIC (500 μM for strains 020001 and 020026, and 250 μM for strain 020032). Iron sources (10 μl) including 10 mg/ml bovine serum albumin (BSA), 1 mM $FeCl_2$, 10 μM hemin, 1 mg/ml hemoglobin, 10 mg/ml holo-transferrin, and 10 mg/ml lactoferrin (Sigma) were spotted directly onto the plate and were incubated 48–72 h at 37 °C. The growth of bacteria was detected by visual inspection.

**Head to head competitions and relative fitness determination.** The relative fitness ($w$) of strain 020026 compared with strains 020001 and 020032 was determined in 24-h head to head competitions in LB broth as described previously[59]. Briefly, the competitors were preconditioned in prewarmed LB broth for 24 h. After that, each strain cultures were measured to the 0.5 McFarland standard and a 10-μl aliquot of each competitor was mixed at a 1:1 ratio. The initial inoculum density of each competitor was approximately $2 \times 10^3$ cfu/ml. The mixture was then inoculated in 10 ml LB broth for 24 h at 37 °C and 200 r.p.m. Strain 020026 was resistant to aztreonam (MIC, >256 μg/ml), while strain 020001 was intermediate to aztreonam (MIC, 8 μg/ml; Supplementary Dataset 1). Therefore, strain 020026 could be differentiated from strain 020001 on agar plates containing 16 μg/ml aztreonam. Strains 020026 and 020032 could be differentiated on agar plates containing 2/4 μg/ml aztreonam–avibactam as strain 020026 was susceptible to aztreonam–avibactam (MIC, 1/4 μg/ml) and strain 020032 was intermediate (MIC, 8/4 μg/ml; Supplementary Dataset 1). Initial ($N_0$) and final ($N_{24}$) densities of each competitor were measured by selective (with 8 μg/ml aztreonam for strains 020026 and 020001 or 2/4 μg/ml aztreonam–avibactam for strains 020026 and 020032) and non-selective (without aztreonam or aztreonam–avibactam) plating on LB agar plates. The $w$ value was calculated using the equation, $w = \log_{10}(Ng_{24}/Ng_0)/\log_{10}(Nw_{24}/Nw_0)$, where $Ng$ and $Nw$ are bacterial densities of strain 020026 and the competitor strain 020001 or 020032, respectively. A $<1w$ value suggests a fitness disadvantage, while $w > 1$ suggests a fitness advantage[60]. Strain 020026 was also competed with strain 020001 in the presence of 375 μM DIP in LB broth and LB agar plates and the competition was performed as described above to determine the relative fitness of strain 020026 under iron-deprived conditions.

**Statistics and reproducibility.** For biofilm formation assays, the differences of absorption values at both OD590 nm and OD595 nm among strains 020026, 020001, and 020032 were compared with one-way ANOVA, which were calculated using SPSS version 21.0 (IBM Analytics; Armonk, NY, USA). For head to head competition, two-tailed *t*-test was used to compare the relative fitness of strain 020026 compared to the competitor strain in non-iron-deprived and iron-deprived conditions, which was calculated using SPSS. All *P* values were two-tailed, and $P < 0.05$ was considered statistically significant.

For biofilm formation assays, all experiments were performed in triplicate (biological replicates) and for each replicate, experiments were repeated nine times (technical replicates). For plasmid stability testing, the experiments for both pNDM5_020026 and pOXA-181 in LB broth and M9 media were all performed in triplicate (biological replicates). All experiments of virulence assay using wax moth were performed with five biological replicates, while iron source growth assay was performed in triplicate (biological replicates). For head to head competition, all experiments were performed in triplicate (biological replicates) and each biological replicate was repeated three times (technical replicates)[61].

**Reporting Summary.** Further information on research design is available in the Nature Research Reporting Summary linked to this article.

## Data availability
Draft genome sequences and short reads of the strains have been deposited in GenBank with the accession numbers being listed in Table 1. The complete sequences of the chromosome and plasmids of strain 020001, 020026, and 020032 have been deposited in GenBank with the accession numbers CP032420 to CP032426, CP034954 to CP034958, and CP034959 to CP034966, respectively. All other data generated or analyzed during this study are included in this article and its supplementary files. Figures 1, 2, 3 and 5 are associated with raw data, which are available as Supplementary datasets. The raw results of Fig. 4 are shown as Supplementary tables.

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

## Acknowledgements

We are grateful for the infection control teams of The Sixth People's Hospital of Chengdu City, Meishan Hospital of Traditional Chinese Medicine, Mianyang Central Hospital, The First People's Hospital of Liangshan Yi Autonomous Prefecture, The Second People's Hospital of Yibin City, The First People's Hospital of Zigong City, and The People's Hospital of Leshan City for collecting the strains and data. We also thank Mrs. Xiaoxia Zhang for performing the virulence assay. This work was supported by grants from the National Natural Science Foundation of China (project no. 81772233, 81661130159, and 8181101395) and the Newton Advanced Fellowship, Royal Society, UK (NA150363).

## Author contributions

Y.F. and Z.Z. designed the study. L.L., J.L., K.M., H.L., and L.W. performed the experiments. Y.F., Y.X., A.M. and Z.Z. analyzed the data. A.M and Z.Z. wrote the paper.

## Additional information

**Competing interests:** The authors declare no competing interests.

