## [Peer Review File · Communications Biology]

Reviewers' comments:

Reviewer #1 (Remarks to the Author):

Review "Key evolutionary events in the emergence of a globally disseminated carbapenem resistant clone in the E coli ST410 lineage"

This manuscript describes a CREC clone from the ST410 lineage of E coli both phenotypically and its genomics.

The main result is that the clone has a single plasmid, but can have either blaOXA181 or blaNDM.

The authors identified specific alleles for certain genes that - could - affect the fitness of the clone.

The authors conclude that the clone is so successful because it acquired certain alleles of genes by recombination.

A. I think the topic of this study is very interesting.

B. I don't understand the main results from the paper, in part because of the writing and in part because figure 1 is very hard to understand. Are the 410 strains from China all B4/H24RxC? If not, how many are? Does this mean that MDR evolved multiple times within ST410? Do they all have the plasmid? Which do and which don't?

C. The paper is very hard to read for someone who doesn't work on bacteria.

1. Lines 62-68 It is not clear if clones 617 and 167 are part of phylogroups A and B1.

2. Lines 68-70 If 410 and 131 are similar because they have an IncX3 plasmid, is this in contrast to 617 and 167?

3. Line 73: ST131 synonymous with ESBL? Are all ESBL strains ST131?

4. Line 78: overlap between 131 and 167, 617: which patterns do you mean?

5. Line 88: Genetic loci which discriminate the MDR clone from the rest of the 410 lineage: I am concerned about the conclusions you draw from this comparison. If the MDR clone arose only once, then any difference between that MDR clone and the others may be due to coincidence (unless the same differences are found, independently in other emerging MDR clones).

6. Line 115: from this I understand that some of the strains that are ST410 from China are part of B4/H24RxC, but some are not, is that correct?

7. Line 110:b Was some of this sequencing was done for another study? Zong et al 2018 Biorxiv? What is the overlap between those two studies, if any?

8. Table 1: I would like it if table 1 would tell me which strains are B4/H24RxC and which are not.

9. Line 130: how many strains were included/excluded?

10. Figure 1: I don't understand which are the Chinese strains here, or which are the B4/H24RxC clone. I don't really understand what I am supposed to see in this figure at all.

11. Line 169: why is this only focused on the B4/H24RxC clone?

12: Line 305: when you say all CREC strains, is it the 25 from China or the 330 worldwide?

13: line 317 Do they all belong to B4/H24RxC? I thought not, but this title suggests that they are?

14: Line 322 and Line 331 is the four strains the same ones?

15: Line 377: You say that there are no SNPs in these genes suggesting that it is not a mutator. But how many SNPs would you have expected? What is the evidence that there must have been recombination.

16: Line 385: This is interesting, that blaNDM was acquired multiple times. I would love to learn more about it. How many times? In how many strains?

Reviewer #2 (Remarks to the Author):

Feng et al., have studied carbapenem-resistant E. coli in Sichuan province, China. They find that many of the CREC are a specific clone, B4/H24RxC, from the ST410 lineage, which has been previously described. Their novel contributions include the identification of B4/H24RxC specific genes and SNPs, and then testing appropriate phenotypes based on these genetic findings to identify biological changes in adhesion and iron metabolism. They also state that blaNDM and blaOXA-181 are frequently interchanged in the same plasmid backbone.

Overall, I think this is an excellent study, I particularly like the genome driven probing of the phenotypes (adherence etc) of the clone. I would also commend the authors for making their WGS data publicly available upon submission. It is a relatively small sample size for a genomic epi study, but the fact they put it into global context makes up for this, along with the detailed analysis and phenotypic studies. The paper is also very well written. I have a few concerns and suggestions, detailed below.

Comments for the Author - major

1. One potential issue with your interpretation is around the 'inter-change' of blaNDM and blaOXA-181 on the IncX3 plasmid. As far as I can see, there is no evidence to differentiate between the following two hypotheses. Is there something that I'm missing which rules out the second one?
 - a. The one you propose – that blaNDM and blaOXA-181 are swapping in and out of the same plasmid backbone.
 - b. That each arrangement arose once, and the plasmid is acquired 'inter-changably'.
2. Line 95-96. What was the selection criteria for the CREC isolates to be included? Was it every CREC identified in the system during this time? If so, please make this clear. If not, please explain the selection criteria.
3. Sequence analysis methods – please confirm that, in addition to extracting SNPs from the freebayes analysis, you also set positions which failed your quality thresholds as 'N' in the consensus genome, rather than leaving them as reference. If you use something like snippy, then this will be done for you.
4. Line 378 to 385 – it is a shame not to have long read sequencing for the strains with pNDM5 and pOXA181. Please investigate the genome graph (FASTG file produced by SPAdes) using Bandage to see whether there is a hint of whether these are on the same or different plasmids. Likewise, please look at the genome graph for the blaNDM carrying isolates which don't have IncX3, to see whether there is an indication from the graph of what kind of plasmid they are associated with. It will be quite difficult because of the large number of plasmids in your samples, but is an interesting technique.
5. Section beginning Line 402 – what is the similarity of the yadC, ybjL and fhuA genes in the three different isolates?
6. I was disappointed by the lack of discussion of the finding that B4/H24RxC has a decreased ability to form biofilms, as this seems to be a counter-intuitive finding. The authors state in the conclusion that this represents the loss of a phenotype which is more associated with environmental survival. However, it seems to me that adhesion to surfaces would still be important in ability to colonise mammalian hosts. Indeed, this review states that loss of adhesive states that loss of adhesive features abolishes long-term colonisation <https://www.ncbi.nlm.nih.gov/pmc/articles/PMC5134996/>.

Please back up your point of view with examples from the literature, or modify your interpretation.

Comments for the Author - minor

1. Line 82, please give the percentage rather than saying 'the vast majority'
2. What library preparation method was used for the MinION sequencing?
3. What yield of long reads did you get? What was the average read length? Etc.
4. Line 769 – 'non-anonymous SNPs' I guess you mean non-synonymous.
5. Line 744 – you say the circular phylogenomic tree is on the left, but it is on the right.
6. Line 747-748 – "Strains belonging to the lineage identified in this study are highlighted in red." It is my understanding that this lineage (B4/H24RxC?) was identified by Roer et al., 2018? It would be helpful to identify B4/H24RxC on the tree.
7. Lines 337-339 – "The mutH, mutL and mutS genes of strain ECS1_14 had no SNPs compared with other strains of this clone, suggesting that the relative high number of SNPs seen in strain ECS1_14 is most likely as a result of recombination rather than due to hypermutation" would you not also be able to see this in the gubbins analysis?

Philip Ashton

Reviewer #3 (Remarks to the Author):

Thank you for the opportunity to review "Key evolutionary events in the emergence of a globally disseminated, carbapenem resistant clone in the Escherichia coli ST410 lineage" by Feng and colleagues. The authors carried out a comprehensive genotypic and phenotypic analysis of an emerging carbapenem resistant clone of E. coli ST410. This included the genomic analysis of 25 recently collected strains in conjunction with 330 published genomes using short and long-read sequencing data and a number of phenotypic assays: plasmid mobility and stability testing, biofilm formation, virulence, iron source growth, and head to head competition. They show that the clone expanded with a single plasmid and that recent adaptations were driven by recombination-acquired alleles in genes important for virulence and survival. They also found frequent switching of the carbapenemase genes blaNDM and blaOXA-181. Overall, the manuscript is clear, cohesive, and well written. The methods are also very well detailed and allow for reproducibility. It demonstrates an excellent example of combining population genomic analysis with microbiology to link genotypic with phenotypic variation to understand how new epidemiologically important clones emerge. As such, their results are appealing to broader audiences than to those just interested in E. coli.

I was particularly interested in the recombination-acquired alleles that resulted in phenotypic differences. The authors posit that these differences are what allowed this clone to emerge as a more relevant clinical pathogen. These assertions were well supported by phenotypic assays. What I felt was partially missing was the timescale of these changes and some conjecture as to the "why". Is there evidence that the alleles were sequentially acquired over a long period of time, or were they the result of recent and frequent recombination events? It appears that the basal branch between ST410 and the other STs is quite long. Does this mean that there is not be enough data to determine the intermediate steps between the evolution of the current dominant clone and other lineages? A coalescent analysis of the clone, using a program like BactDating, would be interesting in the context of their analysis, but it is certainly not needed. Last, I also wonder if these alleles would be identified by gene-wide or site-specific tests of diversifying selection. Is this something the authors considered in their analysis?

My additional comments are as follows:

- 1) When the variation in fhu was first introduced, I was confused as to if the Gubbins analysis supported recent recombination. Based on the results described in line 448, I believe this was the case. Can this be clarified in the text?
- 2) Line 430-431: Were the intergenic SNPs found in promoter regions for genes associated with virulence or survival? This was alluded to but never explicitly discussed.
- 3) Line 57: It may be helpful to spell out KPC, NDM, OXA-48, IMP and VIM, but I will default to whatever is customary in the E. coli literature.
- 4) Line 95: Can the authors provide some additional information about the seven municipal hospitals that the isolates were collected from. At the least, the number of beds would be helpful as well as a sentence about the communities they serve.
- 5) Line 118: Please include the flow cell version for the MinION, library prep kit, and whether the isolates were multiplexed.
- 6) Line 122: Was Pilon run as part of Unicycler or were additional rounds of Pilon used to polish the assemblies? Please clarify and perhaps include the number of iterations Pilon was run on the assemblies.
- 7) Line 133: How many isolates were excluded due to poor coverage?
- 8) Line 166: The Gubbins version number is missing
- 9) Line 440: looks like the text formatting got switched
- 10) Line 136-137: The methods mention the identification/exclusion of phage regions, but this was not described further. Were there phage regions that were excluded?
- 11) Table 1 may be able to be moved to the supplemental since the main pieces of information appear in the text (i.e., collection source and ST)
- 12) Figure 1: It may be clearer to exclude the out-group strain from the clonal phylogeny (red) since the long branch length obscures the topology of the ST410 isolates. Also, even though the methods describe bootstrapping, there are not bootstrap values on the tree. It would be good to include them for the major clades and also include a scale bar for the ST410 phylogeny.
- 13) Table 1: Is there anything that can be inferred from the various sites the strain was isolated from?

Referee expertise:

Referee #1: Evolutionary genetics of drug resistance

Referee #2: Pathogenomics

Referee #3: Bacterial pathogen genomics and genomic epidemiology

Reviewers' comments:

Reviewer #1 (Remarks to the Author):

Review "Key evolutionary events in the emergence of a globally disseminated carbapenem resistant clone in the E coli ST410 lineage"

This manuscript describes a CREC clone from the ST410 lineage of E coli both phenotypically and its genomics.

The main result is that the clone has a single plasmid, but can have either blaOXA181 or blaNDM.

The authors identified specific alleles for certain genes that - could - affect the fitness of the clone.

The authors conclude that the clone is so successful because it acquired certain alleles of genes by recombination.

A. I think the topic of this study is very interesting.

B. I don't understand the main results from the paper, in part because of the writing and in part because figure 1 is very hard to understand. Are the 410 strains from China all B4/H24RxC? If not, how many are? Does this mean that MDR evolved multiple times within ST410? Do they all have the plasmid? Which do and which don't?

C. The paper is very hard to read for someone who doesn't work on bacteria.

1. Lines 62-68 It is not clear if clones 617 and 167 are part of phylogroups A and B1.

Response: Strains of ST167 and ST617 belonged to phylogenetic group A. This information has been added into the revised version.

2. Lines 68-70 If 410 and 131 are similar because they have an IncX3 plasmid, is this in contrast to 617 and 167?

Response: Thank for pointing out this issue. We have reworded the sentence to make the message clearer in the revised version.

3. Line 73: ST131 synonymous with ESBL? Are all ESBL strains ST131?

Response: We have removed “synonymous with the global expansion of ESBL *E. coli*” from the revised version.

4. Line 78: overlap between 131 and 167, 617: which patterns do you mean?

Response: The overlap includes mutations involved in host colonization and in intergenic regions associated with emergence of multi-drug resistant (MDR) plasmid-bearing clones. The sentence has been modified accordingly.

5. Line 88: Genetic loci which discriminate the MDR clone from the rest of the 410 lineage: I am concerned about the conclusions you draw from this comparison. If the MDR clone arose only once, then any difference between that MDR clone and the others may be due to coincidence (unless the same differences are found, independently in other emerging MDR clones).

Response: Thank you for raising this point. The clone has only arisen once, as in ST131, and what we show are unique mutations and gene alleles which have swept to fixation during that clonal emergence. However, we do show parallelism in our results between the events leading to emergence of the B4/H24RxC clone and ST167/617 and ST131 clade C.

6. Line 115: from this I understand that some of the strains that are ST410 from China are part of B4/H24RxC, but some are not, is that correct?

Response: Yes. It is correct.

7. Line 110:b Was some of this sequencing was done for another study? Zong et al 2018 Biorxiv? What is the overlap between those two studies, if any?

Response: No. All CR *E. coli* strains (n=25) recovered in this study were sequenced in this study. Strains from our previous study (Zong et al 2018 Biorxiv) were included for comparison only.

8. Table 1: I would like it if table 1 would tell me which strains are B4/H24RxC and which are not.

Response: The information has been added in Table. Strains of B4/H24RxC are highlighted in bold.

9. Line 130: how many strains were included/excluded?

Response: Six strains. The information has been added into the revised version.

10. Figure 1: I don't understand which are the Chinese strains here, or which

are the B4/H24RxC clone. I don't really understand what I am supposed to see in this figure at all.

Response: The figure has been modified also according to another reviewer's suggestions. There are six ST410 strains from this study, which are highlighted in orange in the figure. Strains of the B4/H24RxC clone, which have a worldwide distribution, are highlighted in the shaded area.

11. Line 169: why is this only focused on the B4/H24RxC clone?

Response: The B4/H24RxC clone is the major ST410 clone that has a worldwide distribution and mediates the spread of carbapenem resistance. In light of its importance, we focused on this clone to investigate the mechanisms underpinning the evolutionary emergence of the clone.

12: Line 305: when you say all CREC strains, is it the 25 from China or the 330 worldwide?

Response: It refers to the 25 strains in this strain. The information has been added into the revised version.

13: line 317 Do they all belong to B4/H24RxC? I thought not, but this title suggests that they are?

Response: Only four of the six Sichuan ST410 strains belong to the B4/H24RxC clone. The title has been modified as suggested.

14: Line 322 and Line 331 is the four strains the same ones?

Response: Yes. They are the same four strains. The strain names have been added in the lines to make the message clearer.

15: Line 377: You say that there are no SNPs in these genes suggesting that it is not a mutator. But how many SNPs would you have expected? What is the evidence that there must have been recombination.

Response: Mutations in these genes are known to cause hypermutations. As there are no SNPs in these genes, the high number of SNPs seen in the strain may not be due to hypermutation. We have modified the wording of the sentence.

16: Line 385: This is interesting, that blaNDM was acquired multiple times. I would love to learn more about it. How many times? In how many strains?

Response: As we state, there are seven strains in the NCBI archives which are NDM positive but do not have IncX3 replicons based on Illumina data available. Nonetheless, we have reworded "multiple times" to "more than once".

Reviewer #2 (Remarks to the Author):

Feng et al., have studied carbapenem-resistant E. coli in Sichuan province,

China. They find that many of the CREC are a specific clone, B4/H24RxC, from the ST410 lineage, which has been previously described. Their novel contributions include the identification of B4/H24RxC specific genes and SNPs, and then testing appropriate phenotypes based on these genetic findings to identify biological changes in adhesion and iron metabolism. They also state that blaNDM and blaOXA-181 are frequently interchanged in the same plasmid backbone.

Overall, I think this is an excellent study, I particularly like the genome driven probing of the phenotypes (adherence etc) of the clone. I would also commend the authors for making their WGS data publicly available upon submission. It is a relatively small sample size for a genomic epi study, but the fact they put it into global context makes up for this, along with the detailed analysis and phenotypic studies. The paper is also very well written. I have a few concerns and suggestions, detailed below.

Comments for the Author - major

1. One potential issue with your interpretation is around the 'inter-change' of blaNDM and blaOXA-181 on the IncX3 plasmid. As far as I can see, there is no evidence to differentiate between the following two hypotheses. Is there something that I'm missing which rules out the second one?

a. The one you propose – that blaNDM and blaOXA-181 are swapping in and out of the same plasmid backbone.

b. That each arrangement arose once, and the plasmid is acquired 'inter-changably'.

Response: This is an excellent point and we have included this in the discussion of the revised version.

2. Line 95-96. What was the selection criteria for the CREC isolates to be included? Was it every CREC identified in the system during this time? If so, please make this clear. If not, please explain the selection criteria.

Response: All CREC in the study period were included. The sentence has been modified in the revised version.

3. Sequence analysis methods – please confirm that, in addition to extracting SNPs from the freebayes analysis, you also set positions which failed your quality thresholds as 'N' in the consensus genome, rather than leaving them as reference. If you use something like snippy, then this will be done for you.

Response: We confirmed that "N" rather than reference sequence was used for sites which failed the quality control. We have re-performed the analysis using the more stringent approach of SNP calling by snippy and the method section has been revised correspondingly.

4. Line 378 to 385 – it is a shame not to have long read sequencing for the strains with PNDM5 and pOXA181. Please investigate the genome graph (FASTG file produced by SPAdes) using Bandage to see whether there is a hint of whether these are on the same or different plasmids. Likewise, please look at the genome graph for the bla_{NDM} carrying isolates which don't have IncX3, to see whether there is an indication from the graph of what kind of plasmid they are associated with. It will be quite difficult because of the large number of plasmids in your samples, but is an interesting technique.

Response: We acknowledge that this is an excellent point. However, the strains are not from this study but from the Philippines and are not available for further sequencing.

We have attempted Bandage as suggested. Assembly graphs in GFA format provided by programs SPAdes and Unicycler were examined carefully to trace the association of replicons and resistance genes. However, due to the nature of short reads and the presence of multiple copies of the same mobile genetic elements such as IS26, ambiguous paths were observed when reaching contigs bounded by IS26 and resulted in an unresolved backbone-resistance relationship. We also used other techniques, such as using BLASTn to identify contigs sharing contents with the reference plasmids pOXA181 and pNDM5_020026 but failed to draw a clear conclusion.

Nonetheless, there were 11 closely related strains from the Philippines in the clade with different resistance gene profiles, i.e., carrying bla_{NDM-1} alone, carrying both bla_{NDM-1} and bla_{OXA-181}, and carrying neither bla_{NDM-1} nor bla_{OXA-181}. This therefore allows us to deduce the likely location of bla_{NDM-1} and bla_{OXA-181} in the five strains carrying both genes by comparing and contrasting the predicted plasmid replicons of the strains within the clade. By this approach, it is likely that bla_{NDM-1} was carried on an IncA/C plasmid, while bla_{OXA-181} was carried on an IncX3 plasmid, in the five strains carrying both bla_{NDM-1} and bla_{OXA-181}. This message has been added into the revised version.

5. Section beginning Line 402 – what is the similarity of the yadC, ybjL and fhuA genes in the three different isolates?

Response: The similarity information has been added into the revised version.

6. I was disappointed by the lack of discussion of the finding that B4/H24RxC has a decreased ability to form biofilms, as this seems to be a counter-intuitive finding. The authors state in the conclusion that this represents the loss of a phenotype which is more associated with environmental survival. However, it seems to me that adhesion to surfaces would still be important in ability to colonise mammalian hosts. Indeed, this review states that loss of adhesive states that loss of adhesive features abolishes long-term colonisation

<https://www.ncbi.nlm.nih.gov/pmc/articles/PMC5134996/>. Please back up your point of view with examples from the literature, or modify your interpretation.

Response: We appreciate the comment. The biofilm formation assay was performed on abiotic surfaces. Although the clone has a decreased ability to form biofilms on abiotic surfaces when compared to other ST410 strains, strains of the clone still have the ability to form biofilms and therefore do not lose the adhesive states. We have added several sentences to the discussion in the revised version as suggested.

Comments for the Author - minor

1. Line 82, please give the percentage rather than saying 'the vast majority'

Response: The percentage has been added into the revised version.

2. What library preparation method was used for the MinION sequencing?

Response: Libraries were constructed using kit SQK-LSK109 and multiplexed using native barcodes from kit EXP-NBD104, according to protocols advised by the manufacturer. The information has been added into the revised version.

3. What yield of long reads did you get? What was the average read length? Etc.

Response: The yield of MinION reads of strain 020001, strain 020026 and strain 020032 is as below, which has been added as Table S4 in the Supplementary material in the revised version.

Strain	020001	020026	020032
Mean read length (bp)	6,633.3	8,124.1	3,752.7
Mean read quality (Q)	10	9.9	9.9
Number of reads	25,399	119,559	226,388
Read length N50 (bp)	8,288	9,710	5,104
Total bases (bp)	168,479,767	971,305,509	849,565,943
> Q7	100%	100%	100%
> Q10	51.4%	49.4%	52.2%

5. Line 744 – you say the circular phylogenomic tree is on the left, but it is on the right.

Response: This has been corrected as suggested.

6. Line 747-748 – “Strains belonging to the lineage identified in this study are highlighted in red.” It is my understanding that this lineage (B4/H24RxC?) was identified by Roer et al., 2018? It would be helpful to identify B4/H24RxC on the tree.

Response: This has been corrected as suggested.

7. Lines 337-339 – “The mutH, mutL and mutS genes of strain ECS1_14 had

no SNPs compared with other strains of this clone, suggesting that the relative high number of SNPs seen in strain ECS1_14 is most likely as a result of recombination rather than due to hypermutation” would you not also be able to see this in the gubbins analysis?

Response: The high number of SNPs seen in the strain was indeed due to recombination, which has been identified by Gubbins. The sentence has been revised accordingly.

Reviewer #3 (Remarks to the Author):

Thank you for the opportunity to review “Key evolutionary events in the emergence of a globally disseminated, carbapenem resistant clone in the *Escherichia coli* ST410 lineage” by Feng and colleagues. The authors carried out a comprehensive genotypic and phenotypic analysis of an emerging carbapenem resistant clone of *E. coli* ST410. This included the genomic analysis of 25 recently collected strains in conjunction with 330 published genomes using short and long-read sequencing data and a number of phenotypic assays: plasmid mobility and stability testing, biofilm formation, virulence, iron source growth, and head to head competition. They show that the clone expanded with a single plasmid and that recent adaptations were driven by recombination-acquired alleles in genes important for virulence and survival. They also found frequent switching of the carbapenemase genes blaNDM and blaOXA-181. Overall, the manuscript is clear, cohesive, and well written. The methods are also very well detailed and allow for reproducibility. It demonstrates an excellent example of combining population genomic analysis with microbiology to link genotypic with phenotypic variation to understand how new epidemiologically important clones emerge. As such, their results are appealing to broader audiences than to those just interested in *E. coli*.

I was particularly interested in the recombination-acquired alleles that resulted in phenotypic differences. The authors posit that these differences are what allowed this clone to emerge as a more relevant clinical pathogen. These assertions were well supported by phenotypic assays. What I felt was partially missing was the timescale of these changes and some conjecture as to the “why”. Is there evidence that the alleles were sequentially acquired over a long period of time, or were they the result of recent and frequent recombination events? It appears that the basal branch between ST410 and the other STs is quite long. Does this mean that there is not be enough data to determine the intermediate steps between the evolution of the current dominant clone and other lineages? A coalescent analysis of the clone, using a program like BactDating, would be interesting in the context of their analysis, but it is certainly not needed. Last, I also wonder if these alleles would be identified by gene-wide or site-specific tests of diversifying selection. Is this something the authors considered in their analysis?

Response: The alleles were present in all genomes of the B4/H24RxC MDR clone, while ST410 strains other than the B4/H24RxC MDR clone have different alleles. This suggests that recombination events to form the alleles occurred at the emergence of the B4/H24RxC MDR clone and have swept to fixation in the clone. This message has been added into the revised version. We have also performed the BactDating analysis. This suggests that the ST410 lineage, and the ancestral branch (sub-lineage of ST410) containing the clone, and the clone emerged > 100 years ago, in September 1994 and in June 2009 respectively. Our analysis revealed that the MDR clone emerged in past recent decade. This analysis has been included into the revised version.

As for the analysis on diversifying section, the reviewer raises an interesting question. We have not pursued this, as the majority of the MDR clone defining SNPs occur in what we postulate to be recombination events as shown in Fig. S in the Supplementary file, making any meaningful selection tests difficult to disentangle from the recombination event, or indeed in which host the selection would have occurred.

My additional comments are as follows:

1) When the variation in *fhu* was first introduced, I was confused as to if the Gubbins analysis supported recent recombination. Based on the results described in line 448, I believe this was the case. Can this be clarified in the text?

Response: Result of Gubbins showed that nucleotide region (between position 3,946,928 and 4,074,660) containing gene *fhu* underwent a recent recombination. This has been added into the revised version.

2) Line 430-431: Were the intergenic SNPs found in promoter regions for genes associated with virulence or survival? This was alluded to by never explicitly discussed.

Response: All identified clone specific SNPs (n=12) in nontranslated intergenic regions (IGRs) belong to the recombination block 3946928 to 4074660. Among which, 5 sites were predicted to be in the -10, -35 boxes of promoter or in 5' UTR of downstream gene, using BPPROM (Solovyev V., Salamov A., 2011). These genes are transcription regulator genes *mraZ* and *rapA*, recombination-promoting family gene *rpn* as well as L-arabinose metabolic gene *araD*. The information has been added into the revised version.

3) Line 57: It may be helpful spell out KPC, NDM, OXA-48, IMP and VIM, but I will default to whatever is customary in the E. coli literature.

Response: This has been modified as suggested.

4) Line 95: Can the authors provide some additional information about the

seven municipal hospitals that the isolates were collected from. At the least, the number of beds would be helpful as well as a sentence about the communities they serve.

Response: The information has been added as a supplementary Table in the revised version.

5) Line 118: Please include the flow cell version for the MinION, library prep kit, and whether the isolates were multiplexed.

Response: The information has been added into the revised version.

6) Line 122: Was Pilon run as part of Unicycler or were additional rounds of Pilon used to polish the assemblies? Please clarify and perhaps include the number of iterations Pilon was run on the assemblies.

Response: For hybrid assemblies, additional rounds (7 on average) of polishing using Pilon were performed apart from the integrated polishing steps in Unicycler, to ensure no change of any type was found. The information has been added into the revised version.

7) Line 133: How many isolates were excluded due to poor coverage?

Response: Six of the 330 strains were excluded. This information has been added into the revised version.

8) Line 166: The Gubbins version number is missing

Response: This has been added into the revised version.

9) Line 440: looks like the text formatting got switched

Response: This has been revised.

10) Line 136-137: The methods mention the identification/exclusion of phage regions, but this was not described further. Were there phage regions that were excluded?

Response: A total of 6 possible prophage regions including 2 intact, 3 incomplete and 1 questionable were identified using PASTER. The results have been shown in a table (Table S5) in the Supplementary file. SNPs within the two intact phage regions, position 1,954,948 to 1,993,973 and position 3,818,437 to 3,854,872, were excluded from all other analyses. The information has been added into the revised version.

11) Table 1 may be able to be moved to the supplemental since the main pieces of information appear in the text (i.e., collection source and ST)

Response: Thank you for the suggestion. We still believe that Table 1 provides important information and it is better to include Table 1 in the manuscript.

12) Figure 1: It may be clearer to exclude the out-group strain from the clonal

phylogeny (red) since the long branch length obscures the topology of the ST410 isolates. Also, even though the methods describe bootstrapping, there are not bootstrap values on the tree. It would be good to include them for the major clades and also include a scale bar for the ST410 phylogeny.

Response: Figure 1 has been reconstructed as suggested to exclude the out-group strain and to add bootstrap values and a scale bar.

13) Table 1: Is there anything that can be inferred from the various sites the strain was isolated from?

Response: This suggests that CREC is associated with various types of infections, which has been added into the revised version.

REVIEWERS' COMMENTS:

Reviewer #2 (Remarks to the Author):

Thank you for revising your manuscript in line with my suggestions.

Reviewer #3 (Remarks to the Author):

Thank you for opportunity to review the revised version of the manuscript. The authors have sufficiently addressed all of my comments. In addition, they have included suggested analysis. Together, these edits have significantly strengthened the manuscript. In particular, the coalescent analysis has provided an evolutionary timeline for the genomic events they detail in the text. I have no additional comments at this time.